# Sketched Lanczos uncertainty score: a low-memory summary of the Fisher information

**Marco Miani**[*]
Technical University of Denmark
mmia@dtu.dk

**Lorenzo Beretta**[*]
Unaffiliated
lorenzo2beretta@gmail.com

**Søren Hauberg**
Technical University of Denmark
sohau@dtu.dk

## Abstract

Current uncertainty quantification is memory and compute expensive, which hinders practical uptake. To counter, we develop SKETCHED LANCZOS UNCERTAINTY (SLU): an architecture-agnostic uncertainty score that can be applied to pre-trained neural networks with minimal overhead. Importantly, the memory use of SLU only grows *logarithmically* with the number of model parameters. We combine Lanczos' algorithm with dimensionality reduction techniques to compute a sketch of the leading eigenvectors of a matrix. Applying this novel algorithm to the Fisher information matrix yields a cheap and reliable uncertainty score. Empirically, SLU yields well-calibrated uncertainties, reliably detects out-of-distribution examples, and consistently outperforms existing methods in the low-memory regime.

## 1 Introduction

The best-performing uncertainty quantification methods share the same problem: *scaling*. Practically this prevents their use for deep neural networks with high parameter counts. Perhaps, the simplest way of defining such a score is to independently train several models, perform inference, and check how consistent predictions are across models. The overhead of the resulting 'Deep Ensemble' [Lakshminarayanan et al., 2017] notably introduces a multiplicative overhead equal to the ensemble size. Current approaches aim to reduce the growth in training time costs by quantifying uncertainty through *local* information of a single pre-trained model. This approach has shown some success for methods like Laplace's approximation [MacKay, 1992, Immer et al., 2021, Khan et al., 2019], SWAG [Maddox et al., 2019], SCOD [Sharma et al., 2021] or Local Ensembles [Madras et al., 2019]. They avoid the need to re-train but still have impractical memory needs.

A popular approach to characterizing local information is the empirical Fisher information matrix, which essentially coincides with the Generalized Gauss-Newton (GGN) matrix [Kunstner et al., 2019]. Unfortunately, for a $p$-parameter model, the GGN is a $p \times p$ matrix, yielding such high memory costs that it cannot be instantiated for anything but the simplest models. The GGN is, thus, mostly explored through approximations, e.g. block-diagonal [Botev et al., 2017], Kronecker-factorized [Ritter et al., 2018a, Lee et al., 2020, Martens and Grosse, 2015] or even diagonal [Ritter et al., 2018b, Miani et al., 2022]. An alternative heuristic is to only assign uncertainties to a subset of the model parameters [Daxberger et al., 2021b, Kristiadi et al., 2020], e.g. the last layer.

Instead, we approximate the GGN with a low-rank matrix. A rank-$k$ approximation of the GGN can be computed using Lanczos algorithm [Madras et al., 2019, Daxberger et al., 2021a] or truncated

---

[*]Equal contribution

38th Conference on Neural Information Processing Systems (NeurIPS 2024).

singular value decomposition (SVD) [Sharma et al., 2021]. These approaches deliver promising uncertainty scores but are limited by their memory footprint. Indeed all aforementioned techniques require $k \cdot p$ memory. Models with high parameter counts are, thus, reduced to only being able to consider *very* small approximate ranks.

**In this work**, we design a novel algorithm to compute the local ensemble uncertainty estimation score introduced by Madras et al. [2019], reintroduced in Section 2.1. Our algorithm is substantially more memory-efficient than the previous one both in theory and practice, thus circumventing the main bottleneck of vanilla Lanczos and randomized SVD (Figure 1). To that end, we employ sketching dimensionality-reduction techniques, reintroduced in Section 2.3, that trade a small-with-high-probability error in some matrix-vector operations for a lower memory usage. Combining the latter with the Lanczos algorithm (reintroduced in Section 2.2) results in the novel SKETCHED LANCZOS. This essentially drops the memory consumption from $\mathcal{O}(pk)$ to $\mathcal{O}(k^2\varepsilon^{-2})$ in exchange for a provably bounded error $\varepsilon$, *independently* on the number of parameters $p$ (up to log-terms).

Applying this algorithm in the deep neural networks settings allows us to scale up the approach from Madras et al. [2019] and obtain a better uncertainty score for a fixed memory budget.

Our contribution is twofold: (1) we prove that orthogonalization approximately commutes with sketching in Section 3, which makes it possible to sketch Lanczos vectors on the fly and orthogonalize them post-hoc, with significant memory savings; (2) we empirically show that, in the low-memory-budget regime, the disadvantage of introducing noise through sketching is outweighed by a higher-rank approximation, thus performing better than baselines when the same amount of memory is used.

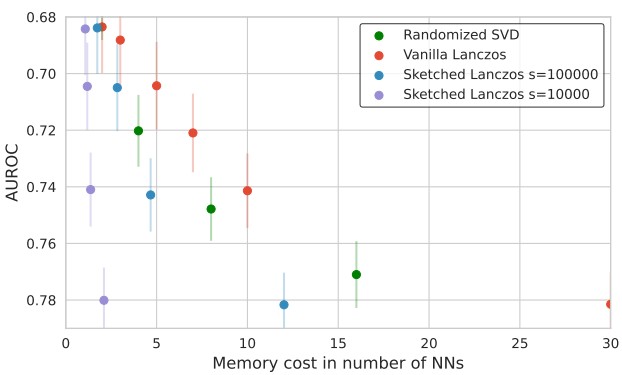

Figure 1: OoD detection performance (↗) on a ResNet.

## 2 Background

Let $f_\theta : \mathbb{R}^d \to \mathbb{R}^t$ denote a neural network with parameter $\theta \in \mathbb{R}^p$, or equivalently $f : \mathbb{R}^p \times \mathbb{R}^d \to \mathbb{R}^t$. Let $\mathbf{J}_{\theta'}(x) = \nabla_\theta f_\theta(x)|_{\theta=\theta'} \in \mathbb{R}^{t\times p}$ be its Jacobian with respect to the parameters, evaluated at a datapoint $x \in \mathbb{R}^d$. Given a training dataset $\mathcal{D} = \{(x_i, y_i)\}_{i=1,\dots,n}$ and a loss function $\mathcal{L}(\theta) = \sum_{(x,y)\in\mathcal{D}} \ell(y, f(x, \theta))$, the Generalized Gauss-Newton matrix (GGN) is defined as

$$\mathbf{G}_\theta = \sum_{i=1}^n \mathbf{J}_\theta(x_i)^\top \mathbf{H}(x_i) \mathbf{J}_\theta(x_i), \tag{1}$$

where $\mathbf{H}(x_i) = \nabla^2_{f_\theta(x_i)} \ell(y_i | f_\theta(x_i)) \in \mathbb{R}^{t\times t}$ is the Hessian of the loss with respect to the neural network output. We reduce the notational load by stacking the per-datum Jacobians into $\mathbf{J}_\theta = [\mathbf{J}_\theta(x_1); \dots; \mathbf{J}_\theta(x_n)] \in \mathbb{R}^{nt\times p}$ and similarly for the Hessians, and write the GGN matrix as $\mathbf{G}_\theta = \mathbf{J}_\theta^\top \mathbf{H} \mathbf{J}_\theta$. For extended derivations and connections with the Fisher matrix we refer to the excellent review by Kunstner et al. [2019]. In the following, we assume access to a pre-trained model with parameter $\theta^*$ and omit the dependency of $\mathbf{G}$ on $\theta^*$.

**Computationally** we emphasize that Jacobian-vector products can be performed efficiently when $f$ is a deep NN. Consequently, the GGN-vector product has twice the cost of a gradient backpropagation, at least for common choices of $\ell$ like MSE or cross-entropy [Khan and Rue, 2021].

### 2.1 Uncertainty score

We measure the uncertainty at a datapoint $x$ as the variance of the prediction $f_\theta(x)$ with respect to a distribution over parameter $\theta$ defined at training time and independently of $x$. This general scheme has received significant attention. For example, Deep Ensemble [Lakshminarayanan et al., 2017] uses a sum of delta distribution supported on independently trained models, while methods that only train

a single network $\theta^*$ generally use a Gaussian $\mathcal{N}(\theta|\theta^*, M)$ with covariance $M \in \mathbb{R}^{p \times p}$. In the latter case, a first-order approximation of the prediction variance is given by a $M$-norm of the Jacobian

$$\text{VAR}_{\theta \sim \mathcal{N}(\theta|\theta^*, M)}[f_\theta(x)] \approx \text{VAR}_{\theta \sim \mathcal{N}(\theta|\theta^*, M)}\left[f_\theta^L(x)\right] = \text{TR}(\mathbf{J}_{\theta^*}(x) \cdot M \cdot \mathbf{J}_{\theta^*}(x)^\top), \quad (2)$$

where $f_\theta^L(x) = f_\theta(x) + \mathbf{J}_{\theta^*}(x) \cdot (\theta^* - \theta)$ is a linearization of $\theta \mapsto f_\theta(x)$ around $\theta^*$.

The GGN matrix (or empirical Fisher; Kunstner et al. [2019]) is notably connected to uncertainty measures and, more specifically, to the choice of the matrix $M$. Different theoretical reasoning leads to different choices and we focus on two of them:

$$M_{\text{eig}} = (\mathbf{G} + \alpha \mathbb{I}_p)^{-1} \qquad\qquad M_{\text{noeig}} = \mathbb{I} - \Pi_{\mathbf{G}}, \qquad (3)$$

where $\Pi_{\mathbf{G}}$ is the projection onto the non-zero eigenvectors of $\mathbf{G}$ and $\alpha > 0$ is a constant.

Madras et al. [2019] justify $M_{\text{noeig}}$ through small perturbation along zero-curvature directions. [2] Immer et al. [2021] justify $M_{\text{eig}}$ in the Bayesian setting where $\alpha$ is interpreted as prior precision. We do not question these score derivations and refer to the original works, but we highlight their similarity. Given an orthonormal eigen-decomposition of $\mathbf{G} = \sum_i \lambda_i v_i v_i^\top$ we see that

$$M_{\text{eig}} = \sum_i \frac{1}{\lambda_i + \alpha} v_i v_i^\top \qquad M_{\text{noeig}} = \sum_i \delta_{\{\lambda_i = 0\}} v_i v_i^\top. \qquad (4)$$

Thus both covariances are higher in the directions of zero-eigenvalues, and the hyperparameter $\alpha$ controls how many eigenvectors are relevant in $M_{\text{eig}}$.

**Practical choices.** The matrix $\mathbf{G}$ is too big to even be stored in memory and approximations are required, commonly in a way that allows access to either the inverse or the projection. A variety of techniques are introduced like diagonal, block diagonal, and block KFAC which also allow for easy access to the inverse. Another line of work, like SWAG [Maddox et al., 2019], directly tries to find an approximation of the covariance $M$ based on stochastic gradient descent trajectories.

Alternatively, low-rank approximations use the eigen-decomposition relative to the top $k$ eigenvalues, which allows direct access to both inverse and projection.

**Is low-rank a good idea?** The spectrum of the GGN has been empirically shown to decay exponentially [Sagun et al., 2017, Papyan, 2018, Ghorbani et al., 2019] (see Figure 2). We investigate this phenomenon further in Appendix C.1 with an ablation over Lanczos hyperparameters. This fast decay implies that the quality of a rank-$k$ approximation of $\mathbf{G}$ improves exponentially w.r.t. $k$ if we measure it with an operator or Frobenius norm, thus supporting the choice of low-rank approximation. Moreover, in the *overparametrized* setting $p \gg nt$, the rank of the GGN is by construction at most $nt$, which is closely linked with functional reparametrizations [Roy et al., 2024].

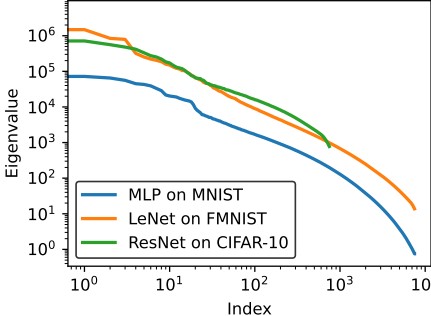

Figure 2: GGN eigenvalues exponential decay. Average and standard deviation over 5 seeds. Details are in Appendix C.1.

The low-rank approach has been applied intensively [Madras et al., 2019, Daxberger et al., 2021a, Sharma et al., 2021] with success, although always limited by memory footprint of $pk$. Madras et al. [2019] argue in favor of a not-too-high $k$ also from a numerical perspective, as using the "small" eigenvectors appears sensitive to noise and consequently is not robust.

## 2.2 The Lanczos algorithm

The Lanczos algorithm is an iterative method for tridiagonalizing a symmetric matrix $G$. If stopped at iteration $k$, Lanczos returns a column-orthonormal matrix $V = [V_1 | \dots | V_k] \in \mathbb{R}^{p \times k}$ and a tridiagonal matrix $T \in \mathbb{R}^{k \times k}$ such that $V^\top G V = T$. The range space of $V$ corresponds to the Krylov subspace $\mathcal{K}_k = \text{span}\{v, Gv, \dots, G^{k-1}v\}$, where $v = V_1$ is a randomly chosen vector. Provably, $\mathcal{K}_k$

---

[2]To be precise, Madras et al. [2019] uses the Hessian rather than the GGN, although their reasoning for discarding the eigenvalues applies to both matrices. Thus, while the score with $M_{\text{noeig}}$ is technically novel, it is a natural connection between Laplace approximations and local ensembles. Also note that some work uses the Hessian in the eigenvalues setting [MacKay, 2003] despite this requires ad-hoc care for negative values.

approximates the eigenspace spanned by the top-$k$ eigenvectors of $G$, i.e. those corresponding to the eigenvalues of largest values [Meurant, 2006]. Thus, $VTV^\top$ approximates the projection of $G$ onto its top-$k$ eigenspace. Notice that projecting $G$ onto its top-$k$ eigenspace yields the best rank-$k$ approximation of $G$ under any unitarily-invariant norm [Mirsky, 1960]. Once the decomposition $G \approx VTV^\top$ is available, we can retrieve an approximation to the top-$k$ eigenpairs of $G$ by diagonalizing $T$ into $T = W\Lambda W^\top$, which can be done efficiently for tridiagonal matrices [Dhillon, 1997]. It has both practically and theoretically been found that this eigenpairs' approximation is very good. We point the reader to Meurant [2006] and Cullum and Willoughby [2002] for a comprehensive survey on this topic.

**The benefits of Lanczos.** Lanczos has two features that make it particularly appealing. First, Lanczos does not need explicit access to the input matrix $G$, but only access to an implementation of $G$-vector product $u \mapsto Gu$. Second, Lanczos uses a small working space: only $3p$ floating point numbers, where the input matrix is $p \times p$. Indeed, we can think of Lanczos as releasing its output in streaming and only storing a small state consisting of the last three vectors $V_{i-1}, V_i$ and $V_{i+1}$.

**The downsides of Lanczos.** Unfortunately, the implementation of Lanczos described above is prone to numerical instability, causing $V_1 \ldots V_k$ to be far from orthogonal. A careful analysis of the rounding errors causing this pathology was carried out by Paige [1971, 1976, 1980]. To counteract, a standard technique is to re-orthogonalize $V_{i+1}$ against all $\{V_j\}_{j \leq i}$, at each iteration. This technique has been employed to compute the low-rank approximation of huge sparse matrices [Simon and Zha, 2000], as well as by Madras et al. [2019] to compute an approximation to the top-$k$ eigenvectors. Unfortunately, this version of Lanczos loses one of the two benefits described above, in that it must store a larger state consisting of the entire matrix $V$. Therefore, we dub this version of the algorithm *hi-memory Lanczos* and the memory-efficient version described above *low-memory Lanczos*. See Appendix C for a comprehensive discussion on these two versions.

**Post-hoc orthogonalization Lanczos.** Alternatively, instead of re-orthogonalizing at every step as in hi-memory Lanczos, we can run low-memory Lanczos, store all the vectors, and orthogonalize all together at the end. Based on the observations of Paige [1980], we expect that orthogonalizing the output of low-memory Lanczos post-hoc should yield an orthonormal basis that approximately spans the top-$k$ eigenspace, similar to hi-memory Lanczos. This post-hoc version of Lanczos is however insufficient. It avoids the cost of orthogonalizing at every iteration but still requires storing the vectors, thus losing again the benefit of memory requirement. Or at least it does unless we find an efficient way to store the vectors.

## 2.3 Sketching

Sketching is a key technique in randomized numerical linear algebra [Martinsson and Tropp, 2020] to reduce memory requirements. Specifically, sketching embeds high dimensional vectors from $\mathbb{R}^p$ into a lower-dimensional space $\mathbb{R}^s$, such that the expected norm error of vector dot-products is bounded, and, as a result, also the score in Equation 2. Here we give a concise introduction to this technique.

**Definition 2.1** (Subspace embedding). Fix $\varepsilon > 0$. A $(1 \pm \varepsilon)$ $\ell_2$-subspace embedding for the column space of an $p \times k$ matrix $U$ is a matrix $S$ for which for all $y \in \mathbb{R}^k$

$$\|SUy\|_2 = (1 \pm \varepsilon)\|Uy\|_2. \tag{5}$$

The goal is to design an *oblivious* subspace embedding, that is a random matrix $S$ such that, for any matrix $U$, $S$ is a subspace embedding for $U$ with sufficiently high probability. In our method, we use a Subsampled Randomized Fourier Transform (SRFT) to achieve this goal [Ailon and Chazelle, 2009]. A SRFT is a $s \times p$ matrix defined by the product $1/\sqrt{sp}PHD$, where $D$ is a diagonal matrix where each diagonal entry is an independent Rademacher random variable, $H$ is the discrete Fourier transform, and $P$ is a diagonal matrix where $s$ random di-

|  | Time | Memory |
|---|---|---|
| Dense JL | $\mathcal{O}(p^\omega)$ | $p^2$ |
| Sparse JL | $\mathcal{O}(p \cdot \varepsilon s)$ | $p \cdot \varepsilon s$ |
| SRFT | $\mathcal{O}(p \log p)$ | $p + s$ |

Table 1: Sketch complexities comparison. Here $\omega$ is such that the current best matrix-multiplication algorithm runs in time $n^\omega$.

agonal entries are set to one and every other entry is set to zero. Thanks to the Fast Fourier Transform algorithm, SRFT can be evaluated in $O(p \log p)$ time, and its memory footprint is only $p + s$.

The following theorem shows that, as long as the sketch size $s$ is big enough, SRFT is an oblivious subspace embedding with high probability.

**Theorem 2.2** (Essentially, Theorem 7 in Woodruff et al. [2014]). *For any $p \times k$ matrix $U$, SRFT is a $(1 \pm \varepsilon)$-subspace embedding for the column space of $U$ with probability $1 - \delta$ as long as $s = \Omega((k + \log p)\varepsilon^{-2} \log(k/\delta))$.*

We stress that, although several other random projections work as subspace embeddings, our choice is not incidental. Indeed other sketches, including the Sparse JL transform [Kane and Nelson, 2014, Nelson and Nguyên, 2013] or the Dense JL transform (Theorem 4, Woodruff et al. [2014]), theoretically have a larger memory footprint or a worse trade-off between $s$ and $k$, as clarified in Table 1. From such a comparison, it is clear that SRFT is best if our goal is to minimize memory footprint. At the same time, evaluation time is still quasilinear.

## 3    Method

We now develop the novel SKETCHED LANCZOS algorithm by combining the 'vanilla' Lanczos algorithm (Section 2.2) with sketching (Section 2.3). Pseudo-code is presented in Algorithm 1. Next, we apply this algorithm in the uncertainty quantification setting and compute an approximation of the score in Equation 2 due to Madras et al. [2019]. Our motivation is that given a fixed memory budget, the much lower memory footprint induced by sketching allows for a higher-rank approximation of $\mathbf{G}$.

### 3.1    Sketched Lanczos

We find the best way to explain our algorithm is to first explain a didactic variant of it, where sketching and orthogonalization happen in reverse order.

Running low-memory Lanczos for $k$ iterations on a $p \times p$ matrix iteratively constructs the columns of a $p \times k$ matrix $V$. Then, post-hoc, we re-orthogonalize the columns of $V$ in a matrix $U \in \mathbb{R}^{p \times k}$. Such a matrix is expensive to store due to the value of $p$, but if we sample a SRFT sketch matrix $S \in \mathbb{R}^{s \times p}$, we can then store a sketched version $SU \in \mathbb{R}^{s \times k}$, saving memory as long as $s < p$. In other words, this is post-hoc orthogonalization Lanczos with a sketching at the end.

We observe that sketching the columns of $U$ is sufficient to $\varepsilon$-preserve the norm of matrix-vector products, with high probability. In particular, the following lemma holds (proof in Appendix A).

**Lemma 3.1** (Sketching low-rank matrices). *Fix $0 < \varepsilon, \delta < 1/2$ and sample a random $s \times p$ SRFT matrix $S$. Then, for any $v \in \mathbb{R}^p$ and any matrix $U \in \mathbb{R}^{p \times k}$ with $\|v\|_2, \|U\|_2 = \mathcal{O}(1)$ we have*

$$\Pr_S \left( \|(SU)^\top (Sv)\|_2 = \|U^\top v\|_2 \pm \varepsilon \right) > 1 - \delta. \tag{6}$$

*as long as $s = \Omega(k\varepsilon^{-2} \cdot \log p \cdot \log(k/\delta))$.*

This algorithm may initially seem appealing since we can compute a tight approximation of $\|U^\top v\|$ by paying only $s \times k$ in memory (plus the neglectable cost of storing $S$). However, as an intermediate step of such an algorithm, we still need to construct the matrix $U$, paying $p \times k$ in memory. Indeed, we defined $U$ as a matrix whose columns are an orthonormal basis of the column space of $V$ and we would like to avoid storing $V$ explicitly and rather sketch each column $V_i$ on the fly, without ever paying $p \times k$ memory. This requires *swapping the order of orthogonalization and sketching*.

This motivates us to prove that if we orthonormalize the columns of $SV$ and apply the same orthonormalization steps to the columns of $V$, then we obtain an approximately orthonormal basis. Essentially, this means that sketching and orthogonalization approximately commute. As a consequence, we can use a matrix whose columns are an orthonormal basis of the column space of $SV$ as a proxy for $SU$ while incurring a small error. Formally, the following holds (proof in Appendix A).

**Lemma 3.2** (Orthogonalizing the sketch). *Fix $0 < \varepsilon, \delta < 1/2$ and sample a random $s \times p$ SRFT matrix $S$. As long as $s = \Omega(k\varepsilon^{-2} \cdot \log p \cdot \log(k/\delta))$ the following holds with probability $1 - \delta$.*

*Given any $p \times k$ full-rank matrix $V$, decompose $V = UR$ and $SV = U_S R_S$ so that $U \in \mathbb{R}^{p \times k}$, $U_S \in \mathbb{R}^{s \times k}$ and both $U$ and $U_S$ have orthonormal columns. For any unit-norm $v \in \mathbb{R}^p$ we have*

$$\|U_S^\top (Sv)\|_2 = \|U^\top v\|_2 \pm \varepsilon. \tag{7}$$

In Lemma 3.1, we proved that given a matrix $U$ with orthonormal columns we can store $SU$ instead of $U$ to compute $v \mapsto \|U^\top v\|_2$ while incurring a small error. However, in our use case, we do

not have explicit access to $U$. Indeed, we abstractly define $U$ as a matrix whose columns are an orthonormal basis of the column space of $V$, but we only compute $U_S$ as a matrix whose columns are an orthonormal basis of the column space of $SV$, without ever paying $p \cdot k$ memory. Lemma 3.2 implies that the resulting error is controlled.

Algorithm 1 lists the pseudocode of our new algorithm: SKETCHED LANCZOS.

---

**Algorithm 1** SKETCHED LANCZOS

---

1: **Input:** Rank $k$, sketch matrix $S \in \mathbb{R}^{s \times p}$, matrix-vector product function $v \mapsto Gv$ for $G \in \mathbb{R}^{p \times p}$.
2: Initialize $v_0 \in \mathbb{R}^p$ as a uniformly random unit-norm vector.
3: **for** $i$ **in** $1, \ldots k$ **do**
4:     $v_i \leftarrow$ LANCZOSITERATION$(v_{i-1}, v \mapsto Gv)$          (glossing over the tridiagonal detail)
5:     Sketch and store $v_i^S \leftarrow Sv_i$
6: **end for**
7: Construct the matrix $V_S = [v_1^S, \ldots, v_k^S] \in \mathbb{R}^{s \times k}$
8: Orthogonalize the columns of $V_S$ and return $U_S \in \mathbb{R}^{s \times k}$

---

**Preconditioned SKETCHED LANCZOS.** We empirically noticed that low-memory Lanczos' stability is quite dependent on the conditioning number of the considered matrix. From this observation, we propose a slight modification of SKETCHED LANCZOS that trades some memory consumption for numerical stability.

The idea is simple, we first run hi-memory Lanczos for $k_0$ iterations, obtaining an approximation of the top-$k_0$ eigenspectrum $U_0 \in \mathbb{R}^{p \times k_0}, \Lambda_0 \in \mathbb{R}^{k_0 \times k_0}$. Then we define a new matrix-vector product

$$\bar{\mathbf{G}}v = (\mathbf{G} - U_0 \Lambda_0 U_0^\top)v \tag{8}$$

and run SKETCHED LANCZOS for $k_1$ iterations on this new, better-conditioned, matrix $\bar{\mathbf{G}}$. This results in a $U_S \in \mathbb{R}^{s \times k_1}$ with sketched orthogonal columns. With $k = k_1 + k_0$, the simple concatenation $[SU_0 | U_S] \in \mathbb{R}^{(k_0+k_1) \times s}$ is a sketched orthogonal $k$-dimensional base of the top-$k$ eigenspace of $\mathbf{G}$, analogous to non-preconditioned SKETCHED LANCZOS. The extra stability comes at a memory cost of $k_0 \cdot p$, thus preventing $k_0$ from being too large.

### 3.2 Sketched Lanczos Uncertainty score (SLU)

The uncertainty score in Equation 2 is computed by first approximating the Generalized Gauss-Newton matrix $\mathbf{G}$. The approach of constructing an orthonormal basis $U \in \mathbb{R}^{p \times k}$ of the top-$k$ eigenvectors of $\mathbf{G}$ with relative eigenvalues $\Lambda$, leads to the low-rank approximation $\mathbf{G} \approx U\Lambda U^\top$. This step is done, for example, by Madras et al. [2019] through hi-memory Lanczos and by Sharma et al. [2021] through truncated randomized SVD. In this step, we employ our novel SKETCHED LANCZOS. Similar to Madras et al. [2019], we focus on the score with $M_{\text{noeig}}$ and thus we neglect the eigenvalues.

Having access to $U$, we can compute the score for a test datapoint $x \in \mathbb{R}^d$ as

$$\text{VAR}[f_\theta(x)] \approx \text{TR}\left(\mathbf{J}_{\theta^*}(x) \cdot (\mathbb{I} - UU^\top) \cdot \mathbf{J}_{\theta^*}(x)^\top\right) = \|\mathbf{J}_{\theta^*}(x)\|_F^2 - \|\mathbf{J}_{\theta^*}(x)U\|_F^2, \tag{9}$$

which clarifies that computing $\|\mathbf{J}_{\theta^*}(x)U\|_F$ is the challenging bit to retrieve the score in Equation 2. Note that $\|\mathbf{J}_{\theta^*}(x)\|_F^2$ can be computed exactly with $t$ Jacobian-vector products.

**Computing the uncertainty score through sketching.** Employing the novel SKETCHED LANCZOS algorithm we can $\varepsilon$-approximate the score in Equation 9, with only minor modifications. Running the algorithm for $k$ iterations returns a matrix $U_S \in \mathbb{R}^{s \times k}$, which we recall is the orthogonalization of $SV$ where $V \in \mathbb{R}^{p \times k}$ are the column vectors iteratively computed by Lanczos.

Having access to $U_S$, we can then compute the score for a test datapoint $x \in \mathbb{R}^d$ as

$$\text{SLU}(x) = \|\mathbf{J}_{\theta^*}(x)\|_F^2 - \|U_S^\top \left(S\mathbf{J}_{\theta^*}(x)^\top\right)\|_F^2 \tag{10}$$

where parentheses indicate the order in which computations should be performed: first a sketch of $\mathbf{J}_{\theta^*}(x)$ is computed, and then it is multiplied by $U_S^\top$. Algorithm 2 summarizes the pipeline.

**Approximation quality.** Recall that the Jacobian $\mathbf{J}_{\theta^*}(x)$ is a $t \times p$ matrix, where $t$ is the output size of the neural network. A slight extension of Lemma 3.2 (formalized in Lemma A.1) implies that the

---
**Algorithm 2** SKETCHED LANCZOS Uncertainty score (SLU)
---
1: **Input:** Observed data $\mathcal{D}$, trained MAP parameter $\theta^* \in \mathbb{R}^p$, approximation rank $k$, sketch size $s$.
2: Initialize SRFT sketch matrix $S \in \mathbb{R}^{s \times p}$
3: Initialize GGN-vector product $v \mapsto \mathbf{G}v = \sum_{x_i \in \mathcal{D}} \mathbf{J}_{\theta^*}(x_i)^\top \mathbf{H}(x_i) \mathbf{J}_{\theta^*}(x_i) v$
4: $U_S \leftarrow$ SKETCHED LANCZOS$(k, S, v \mapsto \mathbf{G}v)$
5: At query time, for a test point $x$ return $\|\mathbf{J}_{\theta^*}(x)\|_F^2 - \|U_S^\top (S\mathbf{J}_{\theta^*}(x)^\top)\|_F^2$
---

score in Equation 9 is guaranteed to be well-approximated by the score in Equation 10 up to a factor $1 \pm \varepsilon$ with probability $1 - \delta$ as long as the sketch size is big enough $s = \Omega(kt\varepsilon^{-2} \log p \log(kt/\delta))$. Neglecting the log terms to develop some intuition, we can think of the sketch size to be $s \approx kt\varepsilon^{-2}$, thus resulting in an orthogonal matrix $U_S$ of size $\approx kt\varepsilon^{-2} \times k$, which also correspond to the memory requirement. From an alternative view, we can expect the error induced by the sketching to scale as $\varepsilon \approx \sqrt{kt/s}$, naturally implying that a larger $k$ will have a larger error, and larger sketch sizes $s$ will have lower error. Importantly, the sketch size $s$ (and consequently the memory consumption and the error bound) depends only logarithmically on the number of parameters $p$, while the memory-saving-ratio $s/p$ clearly improves a lot for bigger architectures.

**Memory footprint.** The memory footprint of our algorithm is at most $4p + s(k + 1)$ floating point numbers. Indeed, the SRFT sketch matrix uses $p + s$ numbers to store $S$, whereas low-memory Lanczos stores at most 3 size-$p$ vectors at a time. Finally, $U_S$ is a $s \times k$ matrix. At query time, we only need to store $S$ and $U_S$, resulting in a memory footprint of $p + s(k + 1)$. Therefore, our method is significantly less memory-intensive than the methods of Madras et al. [2019], Sharma et al. [2021], and all other low-rank Laplace baselines, that use $\Omega(kp)$ memory.

**Time footprint.** The time requirement is comparable to Vanilla Lanczos. We need $k$ SRFT sketch-vector products and each takes $\mathcal{O}(p \log p)$ time, while the orthogonalization of $SV = U_S R$ through QR decomposition takes $\mathcal{O}(pk^2)$ time. Both Vanilla and Sketched algorithm performs $k$ GGN-vector products and each takes $\mathcal{O}(pn)$ time, where $n$ is the size of the dataset, and that is expected to dominate the overall $\mathcal{O}(pk(\log p + k + n))$. Query time is also fast: sketching, Jacobian-vector product and $U_S$-vector product respectively add up to $\mathcal{O}(tp(\log p + 1) + tsk)$.

Note that the linear scaling with output dimension $t$ can slow down inference for generative models. We refer to Immer et al. [2023] for the effect of considering a subset on dataset size $n$ (or on output dimension $t$).

|  | Memory | Time |
|---|---|---|
| Preprocessing | $4p + s(k + 1)$ | $\mathcal{O}(pk(\log p + k + n))$ |
| Query | $p + s(k + 1)$ | $\mathcal{O}(tp(\log p + 1) + tsk)$ |

Table 2: Recall that through the whole paper $p$ is number of parameters, $n$ is dataset size, $t$ is output dimensionality, $k$ is rank approximation and $s$ is sketch size.

## 4 Related work

Modern deep neural networks tend to be more overconfident than their predecessors [Guo et al., 2017]. This motivated intensive research on uncertainty estimation. Here we survey the most relevant work.

Perhaps, the simplest technique to estimate uncertainty over a classification task is to use the softmax probabilities output by the model [Hendrycks and Gimpel, 2016]. A more sophisticated approach, combine softmax with temperature scaling (also Platt scaling) [Liang et al., 2017, Guo et al., 2017]. The main benefit of these techniques is their simplicity: they do not require any computation besides inference. However, they do not extend to regression and do not make use of higher-order information. Moreover, this type of score relies on the extrapolation capabilities of neural networks since they use predictions made far away from the training data. Thus, their poor performance is not surprising.

To alleviate this issue, an immense family of methods has been deployed, all sharing the same common idea of using the predictions of *more than one* model, either explicitly or implicitly. A complete review of these methods is unrealistic, but the most established includes Variational inference [Graves, 2011, Hinton and Van Camp, 1993, Liu and Wang, 2016], Deep ensembles [Lakshminarayanan et al., 2017], Monte Carlo dropout [Gal and Ghahramani, 2016, Kingma et al., 2015] and Bayes by Backprop [Blundell et al., 2015].

More closely related to us, the two uncertainty quantification scores introduced in Equation 2 with covariances $M_{\text{eig}}$ and $M_{\text{noeig}}$ (Equation 3) have been already derived from Bayesian (with Laplace's approximation) and frequentist (with local perturbations) notions of underspecification, respectively:

**Laplace's approximation.** By interpreting the loss function as an unnormalized Bayesian log-posterior distribution over the model parameters and performing a second-order Taylor expansion, Laplace's approximation [MacKay, 1992] results in a Gaussian approximate posterior whose covariance matrix is the loss Hessian. The linearized Laplace approximation [Immer et al., 2021, Khan et al., 2019] further linearize $f_\theta$ at a chosen weight $\theta^*$, i.e. $f_\theta(x) \approx f_{\theta^*}(x) + \mathbf{J}_{\theta^*}(x)(\theta - \theta^*)$. In this setting the posterior is exactly Gaussian and the covariance is exactly $M_{\text{eig}}$.

**Local perturbations.** A different, frequentist, family of methods studies the change in optimal parameter values induced by change in observed data. A consequence is that the parameter directions corresponding to functional invariance on the training set are the best candidates for OoD detection. This was directly formalized by Madras et al. [2019] which derives the score with covariance $M_{\text{noeig}}$, which they call a local ensemble. A similar objective is approximated by Resampling Under Uncertainty (RUE, Schulam and Saria [2019]) perturbing the training data via influence functions, and by Stochastic Weight Averaging Gaussian (SWAG, Maddox et al. [2019]) by following stochastic gradient descent trajectories.

**Sketching in the deep neural networks literature.** Sketching techniques are not new to the deep learning community. Indeed several works used randomized SVD [Halko et al., 2011], which is a popular algorithm [Tropp and Webber, 2023] that leverages sketch matrices to reduce dimensionality and compute an approximate truncated SVD faster and with fewer passes over the original matrix. It was used by Antorán et al. [2023] to compute a preconditioner for conjugate gradient and, similarly, by Mishkin et al. [2018] to extend Variational Online Gauss-Newton (VOGN) training [Khan et al., 2018]. More related to us, Sketching Curvature for OoD Detection (SCOD, Sharma et al. [2021]) uses Randomized SVD to compute exactly the score in Equation 2 with $M_{\text{eig}}$, thus serving as a baseline with an alternative to Lanczos. We compare to it more extensively in Appendix B.

*To the best of our knowledge* no other work in uncertainty estimation for deep neural networks uses sketching directly to reduce the size of the data structure used for uncertainty estimation. Nonetheless, a recent line of work in numerical linear algebra studies how to apply sketching techniques to Krylov methods, like Lanczos or Arnoldi [Balabanov and Grigori, 2022, Timsit et al., 2023, Simoncini and Wang, 2024, Güttel and Schweitzer, 2023]. We believe sketching is a promising technique to be applied in uncertainty estimation and, more broadly, in Bayesian deep learning. Indeed, all the techniques based on approximating the GGN could benefit from dimensionality reduction.

## 5 Experiments

With a focus on memory budget, we benchmark our method against a series of methods, models, and datasets. The code for both training and testing is implemented in JAX [Bradbury et al., 2018] and it is publicly available[3]. Details, hyperparameters and more experiments can be found in Appendix D.

To evaluate the uncertainty score we measure the performance of out-of-distribution (OoD) detection and report the Area Under Receiver Operator Curve (AUROC). We choose **models** with increasing complexity and number of parameters: MLP, LeNet, ResNet, VisualAttentionNet and SwinTransformer architectures, with the number of parameters ranging from 15K to 200M. We train such models on 5 different **datasets**: MNIST [Lecun et al., 1998], FASHIONMNIST [Xiao et al., 2017], CIFAR-10 [Krizhevsky et al., 2009], CELEBA [Liu et al., 2015] and IMAGENET [Deng et al., 2009]. We test the score performance on a series of OoD datasets, including rotations and corruptions [Hendrycks, 2019] of ID datasets, as well as SVHN [Netzer et al., 2011] and FOOD101 [Bossard et al., 2014]. For CELEBA and IMAGENET we hold out some classes from training and use them as OoD datasets.

We compare to the most relevant **methods** in literature: Linearized Laplace Approximation with a low-rank structure (LLA) [Immer et al., 2021] and Local Ensemble (LE) are the most similar to us. We also consider Laplace with diagonal structure (LLA-D) and Local Ensemble Hessian variant (LE-H) [Madras et al., 2019]. Another approach using randomized SVD instead of Lanczos is Sketching Curvature for OoD Detection (SCOD) [Sharma et al., 2021], which also serves as a Lanczos baseline. Lastly, we include SWAG [Maddox et al., 2019] and Deep Ensemble (DE) [Lakshminarayanan et al., 2017].

---

[3] `https://github.com/IlMioFrizzantinoAmabile/uncertainty_quantification`

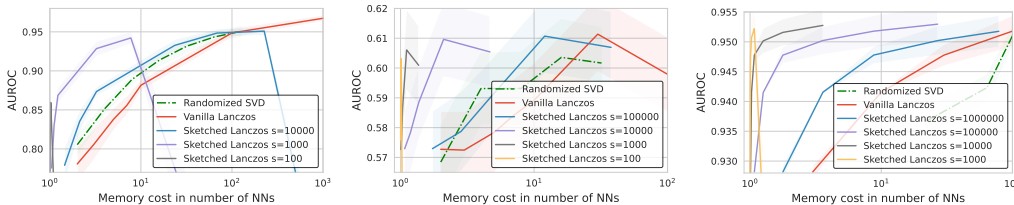

Figure 3: Sketch sizes $s$ comparison for: LeNet $p = 40K$ on FASHIONMNIST vs MNIST (left), ResNet $p = 300K$ on CIFAR-10 vs CIFAR-corrupted with defocus blur (center), and VisualAttentionNet $p = 4M$ on CELEBA vs FOOD101 (right). The lower the ratio $s/p$, the stronger the memory efficiency.

**Effect of different sketch sizes.** We expect the error induced by the sketching to scale as $\epsilon \approx \sqrt{k/s}$, thus a larger $k$ will have a larger error, and larger sketch sizes $s$ will have a lower error, independently on number of parameters up to log-terms. On the other hand, a larger parameter count will have a better memory-saving-ratio $s/p$, leading to an advantage for bigger architectures as shown in Figure 3.

| model | MLP $p = 15K$ | | | LeNet $p = 40K$ | | ResNet $p = 200K$ | | |
|---|---|---|---|---|---|---|---|---|
| ID data | MNIST vs | | | FASHIONMNIST vs | | CIFAR-10 vs | | |
| OoD data | FASHIONMNIST | KMNIST | Rotation (avg) | MNIST | Rotation (avg) | SVHN | CIFAR-100 | Corrupt (avg) |
| SLU (us) | $0.26 \pm 0.02$ | $0.42 \pm 0.04$ | $0.59 \pm 0.02$ | $\mathbf{0.94 \pm 0.01}$ | $0.74 \pm 0.03$ | $\mathbf{0.76 \pm 0.05}$ | $\mathbf{0.54 \pm 0.02}$ | $0.57 \pm 0.04$ |
| LLA | $0.27 \pm 0.03$ | $0.30 \pm 0.03$ | $0.49 \pm 0.01$ | $0.80 \pm 0.03$ | $0.70 \pm 0.03$ | $0.67 \pm 0.07$ | $\mathbf{0.54 \pm 0.02}$ | $0.57 \pm 0.05$ |
| LLA-D | $\mathbf{0.94 \pm 0.03}$ | $\mathbf{0.98 \pm 0.01}$ | $\mathbf{0.73 \pm 0.01}$ | $0.68 \pm 0.07$ | $0.61 \pm 0.05$ | $0.55 \pm 0.10$ | $0.52 \pm 0.02$ | $0.52 \pm 0.04$ |
| LE | $0.27 \pm 0.03$ | $0.30 \pm 0.03$ | $0.49 \pm 0.01$ | $0.80 \pm 0.03$ | $0.70 \pm 0.03$ | $0.67 \pm 0.07$ | $\mathbf{0.54 \pm 0.02}$ | $0.57 \pm 0.05$ |
| LE-H | $0.27 \pm 0.03$ | $0.30 \pm 0.03$ | $0.49 \pm 0.01$ | $0.80 \pm 0.03$ | $0.70 \pm 0.03$ | $0.66 \pm 0.07$ | $\mathbf{0.54 \pm 0.02}$ | $0.57 \pm 0.05$ |
| SCOD | $0.27 \pm 0.03$ | $0.31 \pm 0.03$ | $0.51 \pm 0.01$ | $0.84 \pm 0.02$ | $0.71 \pm 0.03$ | $0.68 \pm 0.07$ | $\mathbf{0.54 \pm 0.02}$ | $\mathbf{0.58 \pm 0.04}$ |
| SWAG | $0.29 \pm 0.05$ | $0.19 \pm 0.02$ | $0.41 \pm 0.03$ | $0.75 \pm 0.02$ | $0.70 \pm 0.04$ | $0.48 \pm 0.11$ | $0.49 \pm 0.01$ | $0.52 \pm 0.06$ |
| DE | $\mathbf{0.94 \pm 0.04}$ | $0.96 \pm 0.00$ | $0.71 \pm 0.02$ | $0.92 \pm 0.01$ | $\mathbf{0.79 \pm 0.02}$ | $0.46 \pm 0.02$ | $0.51 \pm 0.00$ | $0.50 \pm 0.01$ |

| model | VisualAttentionNet $p = 4M$ | | SWIN $p = 200M$ | | | | | | | |
|---|---|---|---|---|---|---|---|---|---|---|
| ID data | CELEBA vs | | IMAGENET vs | | | | | | | |
| OoD data | FOOD-101 | Hold-out (avg) | Castle | Flamingo | Lighter | Odometer | Parachute | Pineapple | Triceratops | Volcano |
| SLU (us) | $\mathbf{0.95 \pm 0.003}$ | $\mathbf{0.72 \pm 0.02}$ | $\mathbf{0.57}$ | $\mathbf{0.64}$ | $\mathbf{0.63}$ | $0.37$ | $\mathbf{0.70}$ | $\mathbf{0.56}$ | $\mathbf{0.69}$ | $\mathbf{0.83}$ |
| LLA | $0.93 \pm 0.001$ | $0.67 \pm 0.02$ | $0.49$ | $0.56$ | $0.59$ | $0.41$ | $0.56$ | $0.45$ | $0.67$ | $0.78$ |
| LLA-D | $0.80 \pm 0.02$ | $0.52 \pm 0.03$ | $0.51$ | $0.54$ | $0.52$ | $0.32$ | $0.45$ | $0.48$ | $0.68$ | $0.70$ |
| LE | $0.93 \pm 0.001$ | $0.67 \pm 0.02$ | $0.51$ | $0.58$ | $0.62$ | $0.41$ | $0.59$ | $0.47$ | $0.67$ | $0.80$ |
| LE-H | $0.91 \pm 0.002$ | $0.64 \pm 0.03$ | $0.50$ | $0.57$ | $0.60$ | $0.41$ | $0.57$ | $0.47$ | $0.67$ | $0.80$ |
| SCOD | $0.94 \pm 0.00$ | $0.68 \pm 0.02$ | na | na | na | na | na | na | na | na |
| SWAG | $0.69 \pm 0.04$ | $0.46 \pm 0.04$ | $0.46$ | $0.54$ | $0.55$ | $\mathbf{0.52}$ | $0.44$ | $0.36$ | $0.64$ | $0.72$ |
| DE | $0.88 \pm 0.02$ | $0.63 \pm 0.02$ | na | na | na | na | na | na | na | na |

Table 3: AUROC scores of SKETCHED LANCZOS Uncertainty vs baselines with memory budget of $3p$. Results for all OoD datasets are illustrated more extensively in Figure 4. Mean values and standard deviations for each table block obtained over 10, 10, 5, 3, 1 independently trained models, respectively.

**Summary of the experiments.** For *most of* the ID-OoD dataset pairs we tested, our SKETCHED LANCZOS Uncertainty outperforms the baselines, as shown in Table 3 and more extensively in Figure 4 where we fix the memory budget to be $3p$. Deep Ensemble performs very well in the small architecture but progressively deteriorates for bigger parameter sizes. SWAG outperforms the other methods on some specific choices of CIFAR-10 corruptions, but we found this method to be extremely dependent on the choice of hyperparameters. SCOD is also a strong baseline in some settings, but we highlight that it requires instantiating the full Jacobian with an actual memory requirement of $tp$. These memory requirements make SCOD inapplicable to ImageNet. In this setting, given the significant training time, also DEEP ENSEMBLE becomes not applicable.

The budget of $3p$ is an arbitrary choice, but the results are consistent with different values. More experiments, including a $10p$ memory budget setting, a study on the effect of preconditioning, and a synthetic-data ablation on the trade-off sketch size vs low rank, are presented in Appendix D.3.

## 6 Conclusion

We have introduced SKETCHED LANCZOS, a powerful memory-efficient technique to compute approximate matrix eigendecompositions. We take a first step in exploiting this technique showing that sketching the top eigenvectors of the Generalized Gauss-Newton matrix leads to high-quality

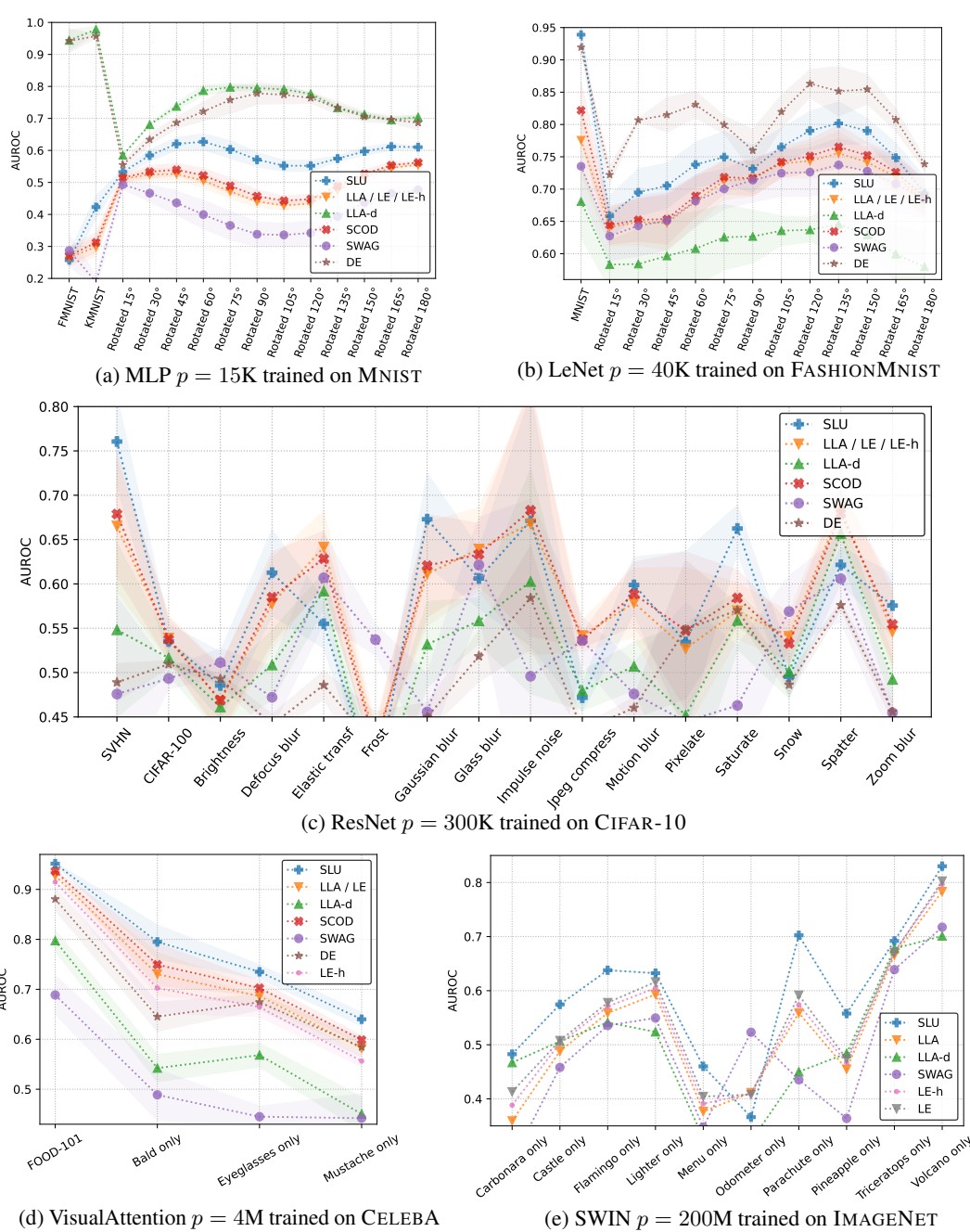

Figure 4: AUROC scores of SKETCHED LANCZOS Uncertainty vs baselines with memory budget $3p$. SLU outperforms the baselines on several choices of ID (4a, 4b, 4c, 4d, 4e) and OoD (x-axis) datasets pairs. Dashed lines are for improved visualization only; see Table 3 for values and standard deviations. Plots 4a, 4b, 4c, 4d, 4e are averaged respectively over 10, 10, 5, 3, 1 independently trained models.

scalable uncertainty measure. We empirically show the superiority of the SKETCHED LANCZOS Uncertainty score (SLU) among a variety of baselines in the low-memory-budget setting, where the assumption is that the network has so many parameters that we can only store a few copies.

**Limitations.** The data structure produced by Sketched Lanczos is sufficiently rich to evaluate the predictive variance, and consequently the uncertainty score. However, from a Bayesian point of view, it is worth noting that the method does not allow us to sample according to the posterior.

## Acknowledgments and Disclosure of Funding

The work was partly funded by the Novo Nordisk Foundation through the Center for Basic Machine Learning Research in Life Science (NNF20OC0062606). It also received funding from the European Research Council (ERC) under the European Union's Horizon program (101125993), and from a research grant (42062) from VILLUM FONDEN. The authors acknowledge the Pioneer Centre for AI, DNRF grant P1. The authors also acknowledge Scuola Normale Superiore of Pisa.

We thank Marcel Schweitzer for pointing out some highly relevant literature on applying sketching to Krylov methods carried out by the numerical linear algebra community.

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

# A  Dimensionality Reduction

## A.1  Proof of "Sketching low-rank matrices" Lemma

In this section, we prove Lemma 3.1. We restate it below for convenience.

**Lemma 3.1** (Sketching low-rank matrices). *Fix $0 < \varepsilon, \delta < 1/2$ and sample a random $s \times p$ SRFT matrix $S$. Then, for any $v \in \mathbb{R}^p$ and any matrix $U \in \mathbb{R}^{p \times k}$ with $||v||_2, ||U||_2 = \mathcal{O}(1)$ we have*

$$\Pr_S \Big( ||(SU)^\top (Sv)||_2 = ||U^\top v||_2 \pm \varepsilon \Big) > 1 - \delta. \tag{6}$$

*as long as $s = \Omega(k\varepsilon^{-2} \cdot \log p \cdot \log(k/\delta))$.*

*Proof.* In order to prove that $||(SU)^\top (Sv)||_2 = ||U^\top v||_2 \pm \varepsilon$ with probability $\geq 1 - \delta$ we prove the following stronger statement. Denote with $[u]_i$ the $i$-th coordinae of $u$, then for each $i = 1 \ldots k$ we have $[(SU)^\top(Sv)]_i = [U^\top v]_i \pm \varepsilon/\sqrt{k}$ with probability at least $1 - \delta/k$. Apparently, the former follows from the latter by union bound.

Notice that $[(SU)^\top(Sv)]_i = (SU^i)^\top(Sv)$, where $U^i$ is the $i$-th column of $U$. Moreover, as long as $s = \Omega(k\varepsilon^{-2} \cdot \log p \cdot \log(k/\delta))$, then $S$ is a $(1 \pm \varepsilon/\sqrt{k})$-subspace embedding for the 2-dimensional subspace spanned by $v$ and $U^i$ with probability $1 - \delta/k$ (Theorem 2.2). Therefore, conditioning on this event:

$$
\begin{aligned}
(SU^i)^\top(Sv) &= \frac{1}{4} \left( ||S(U^i + v)||_2^2 - ||S(U^i - v)||_2^2 \right) \\
&= \frac{1}{4} \left( ||(U^i + v)||_2^2 - ||(U^i - v)||_2^2 \right) \pm O(\varepsilon/\sqrt{k}) \\
&= (U^i)^\top v \pm O(\varepsilon/\sqrt{k}) \\
&= [U^\top v]_i \pm O(\varepsilon/\sqrt{k}).
\end{aligned}
$$

The second equality sign holds because of the subspace embedding property, together with $||U^i + v||_2, ||U^i - v||_2 = O(1)$. $\qquad\square$

## A.2  Proof of "Orthogonalizing the sketch" Lemma

In this section, we prove Lemma 3.2. We restate it below for convenience.

**Lemma 3.2** (Orthogonalizing the sketch). *Fix $0 < \varepsilon, \delta < 1/2$ and sample a random $s \times p$ SRFT matrix $S$. As long as $s = \Omega(k\varepsilon^{-2} \cdot \log p \cdot \log(k/\delta))$ the following holds with probability $1 - \delta$.*

*Given any $p \times k$ full-rank matrix $V$, decompose $V = UR$ and $SV = U_S R_S$ so that $U \in \mathbb{R}^{p \times k}$, $U_S \in \mathbb{R}^{s \times k}$ and both $U$ and $U_S$ have orthonormal columns. For any unit-norm $v \in \mathbb{R}^p$ we have*

$$||U_S^\top (Sv)||_2 = ||U^\top v||_2 \pm \varepsilon. \tag{7}$$

Essentially, the proof of Lemma 3.2 was already known, see for example Corollary 2.2 in Balabanov and Grigori [2022]. Nonetheless, we include a short proof for completeness' sake.

*Proof.* Since $V$ is full-rank, it is easy to verify that $SV$ is also full rank with high probability. Thus, $R$ is non-singular and we can define $\bar{U} = VR_S^{-1}$. Notice that $U_S = S\bar{U}$. Now, we prove that

$$||UU^\top - \bar{U}\bar{U}^\top||_2 \leq \varepsilon \tag{11}$$

where $||\cdot||_2$ is the operator norm. By definition of $U_S$, we have $I_k = U_S^\top U_S = \bar{U}S^\top S\bar{U}^\top$ By subspace embedding property of $S$, we have that

$$||\bar{U}\bar{U}^\top - I_k||_2 = ||\bar{U}\bar{U}^\top - \bar{U}S^\top S\bar{U}^\top||_2 \leq \varepsilon$$

with probability $1 - \delta$. Conditioning on this event, the singular values of $\bar{U}$ lie in $[1 - \varepsilon, 1 + \varepsilon]$. Moreover, $Ran(U) = Ran(\bar{U})$. Therefore, taking the singular value decomposition (SVD) of $U$ and $\bar{U}$ yields Equation 11.

Now we are left to prove that $||\bar{U}^\top v||_2 = ||U_S^\top Sv||_2 \pm \varepsilon$. We have $U_S^\top S = \bar{U}^\top S^\top S$ and by Lemma 3.1 we have $||\bar{U}^\top S^\top Sv||_2 = ||\bar{U}^\top v||_2 \pm \varepsilon$ with probability $1 - \delta$. Thus, $||\bar{U}^\top v||_2 = ||U_S^\top Sv||_2 \pm \varepsilon$ and combined with Equation 11 gives Equation 7, up to constant blow-ups in $\varepsilon$ and $\delta$. $\qquad\square$

### A.3 Extension of Lemma 3.2

In this section we extend Lemma 3.2 to the case where instead of having a query vector $v \in \mathbb{R}^p$ we have a query matrix $J \in \mathbb{R}^{p \times t}$. In our application, $J$ is the Jacobian $\mathbf{J}_{\theta^*}(x)$.

**Lemma A.1** (Orthogonalizing the sketch, for matrix queries). *Fix $0 < \varepsilon, \delta < 1/2$ and sample a random $s \times p$ SRFT matrix $S$. As long as $s = \Omega(tk\varepsilon^{-2} \cdot \log p \cdot \log(tk/\delta))$ the following holds with probability $1 - \delta$.*

*Given any $p \times k$ full-rank matrix $V$, decompose $V = UR$ and $SV = U_S R_S$ so that $U \in \mathbb{R}^{p \times k}$, $U_S \in \mathbb{R}^{s \times k}$ and both $U$ and $U_S$ have orthonormal columns. For any $J \in \mathbb{R}^{p \times t}$ we have*

$$\|J\|_F^2 - \|U_S^\top (SJ)\|_F^2 = (1 \pm \varepsilon)\|J\|_F^2 - \|U^\top J\|_F^2. \tag{12}$$

*Proof.* Let $J^i$ the $i$-th column of $J$. Suppose that the following intermediate statement holds: for each $i$, $\|U_S^\top (SJ^i)\|_2 = \|U^\top J^i\|_2 \pm \|J\|_F \cdot \varepsilon/\sqrt{t}$ with probability $1 - \delta/t$. If the statement above holds, then we have

$$\|\|U_S^\top (SJ)\|_F^2 - \|U^\top J\|_F^2\| =$$
$$(\|U_S^\top (SJ)\|_F + \|U^\top J\|_F) \cdot \|\|U_S^\top (SJ)\|_F - \|U^\top J\|_F\| \leq$$
$$O(\|J\|_F) \cdot \sqrt{\sum_i (\|U_S^\top (SJ^i)\|_2 - \|U^\top J^i\|_2)^2} \leq$$
$$O(\|J\|_F^2 \cdot \varepsilon).$$

The penultimate inequality holds because of triangle inequality, whereas the last inequality holds with probability $1 - \delta$ by union bound.

To prove the intermediate statement above, it is sufficient to apply Lemma 3.2 with $\varepsilon' = \varepsilon/\sqrt{t}$ and $\delta' = \delta/t$. $\qquad\square$

## B  Extended Related Work

**Local ensembles.**  In Madras et al. [2019], they consider *local ensembles*, a computationally efficient way to detect underdetermination without training an actual ensemble of models. A local ensemble is a distribution over models obtained by perturbing the parameter along zero-curvature directions. Small parameter perturbations along zero-curvature directions cause small perturbation of the model prediction. Given a test point, Madras et al. [2019] define the local ensemble *score* as the first-order term of the predictive variance for small perturbations along zero-curvature directions.

Moreover, in Madras et al. [2019] they prove that the aforementioned score is equivalent to $\|\Pi_k \mathbf{J}_{\theta^*}(x)\|_F$ where $\Pi_k$ is a projection onto a subspace defined as the orthogonal of the zero-curvature subspace at $\theta^*$, $\mathbf{J}_{\theta^*}(x)$ is the Jacobian of the test datapoint and $\|\cdot\|_F$ is the Frobenius norm. Computationally, they approximate their score by setting $\Pi_k = I - U_k U_k^\top$, where $U_k$ is a $p \times k$ matrix which columns are the top-$k$ eigenvectors of the Hessian of the loss $\mathcal{L}(\theta)$.

Their method uses Lanczos algorithm to compute the top-$k$ eigenvectors of the Hessian and uses $k \cdot p$ memory. This is costly, indeed Madras et al. [2019] write

> *"The main constraint of our method is space rather than time — while estimating the first $k$ eigenvectors enables easy caching for later use, it may be difficult to work with these eigenvectors in memory as $k$ and model size $p$ increase. [...] we note in some cases that increasing $k$ further could have improved performance [...] This suggests that further work on techniques for mitigating this tradeof."*

**Comparison with SCOD.**  SCOD, the uncertainty estimation score introduced in Sharma et al. [2021], has gained traction lately and it has been used as an uncertainty estimation baseline in applications Qiao et al. [2023], Elhafsi et al. [2023]. SCOD can be interpreted as a Bayesian method that approximates the posterior with a Gaussian having a covariance matrix of the form identity + low-rank. Computing such approximation of the variance boils down to computing a low-rank approximation of the inverse of $F + \alpha I$, where $F$ is the Fisher information matrix and $\alpha$ depends

on the variance of the isotropic Gaussian prior on $\theta$. Ultimately, the SCOD score is defined as $\|(I - U_k \Lambda U_k^\top)\mathbf{J}_{\theta^*}(x)\|_F$, where $U_k$ is a $p \times k$ matrix which columns are the top-$k$ eigenvectors of $F$ and $\Lambda$ is a $k \times k$ diagonal matrix such that $\Lambda_{ii} = \lambda_i/(\lambda_i + \alpha)$.

SCOD and Local Ensembles scores are extremely similar. Indeed, the only difference between their scores is that SCOD weighs projection onto different eigenvectors according to $\Lambda$. However, in Sharma et al. [2021] they observe empirically that different values of $\alpha$ yield the same score accuracy, hinting that replacing $\Lambda$ with the identity (that essentially gives local ensembles score) should have a modest impact on accuracy. From a computational point of view, both methods compute the top-$k$ eigenvectors of the Hessian / Fisher accessing it solely through matrix-vector product. Local ensembles achieves that using Lanczos algorithm, whereas SCOD uses truncated randomized singular values decomposition (SVD). In both cases the space complexity $k \cdot p$. Similarly to Madras et al. [2019], also Sharma et al. [2021] observed that the most prominent limiting factor for their method was memory usage[4]:

> *"The memory footprint of the offline stage of* SCOD *is still linear, i.e., $O(kp)$. As a result, GPU memory constraints can restrict $k$ [...] substantially for large models."*

**GGN / Fisher vs Hessian.**  Local Ensembles uses computes the top-$k$ eigenvalue of the Hessian of $\mathcal{L}(\theta^*)$, whereas Sharma et al. [2021], Immer et al. [2021] employs the Fisher / GGN. Implementing matrix-vector product for both Hessian and Fisher /GGN is straightforward, thanks to the power of modern software for automatic differentiation. However, we believe that using the GGN / Fisher is more appropriate in this context. Indeed, the latter is guaranteed to be positive semi-definite (PSD), whereas the Hessian is not necessarily PSD for non-convex landscapes.

## C  Lanczos algorithm

Lanczos algorithm is an iterative method for tridiagonalizing an Hermitian[5] matrix $G$. If stopped at iteration $k$, Lanczos returns a column-orthogonal matrix $V = [V_1 | \dots | V_k] \in \mathbb{R}^{p \times k}$ and a tridiagonal matrix $T \in \mathbb{R}^{k \times k}$ such that $V^\top G V = T$. The range space of $V$ corresponds to the Krylov subspace $\mathcal{K}_k = span\{v, Gv, \dots, G^{k-1}v\}$, where $v = V_1$ is a randomly chosen vector. Provably $\mathcal{K}_k$ approximates the eigenspace spanned by the top-$k$ eigenvectors (those corresponding to eigenvalues of largest modulus) of $G$. Thus, $VTV^\top$ approximates the projection of $G$ onto its top-$k$ eigenspace. Notice that projecting $G$ onto its top-$k$ eigenspace yields the best rank-$k$ approximation of $G$ under any unitarily-invariant norm Mirsky [1960]. Moreover, as observed in the previous section the spectrum of the GGN decays rapidly, making low-rank decomposition particularly accurate.

Once the decomposition $G \approx VTV^\top$ is available, we can retrieve an approximation to the top-$k$ eigenpairs of $G$ by diagonalizing $T$ into $T = W\Lambda W^\top$, which can be done efficiently for tridiagonal matrices Dhillon [1997]. The quality of eigenpairs' approximation has been studied in theory and practice. We point the reader to Meurant [2006], Cullum and Willoughby [2002] for a comprehensive sourvey on this topic.

**Lanczos, in a nutshell.**  Here we give a minimal description of Lanczos algorithm. Lanczos maintains a set of vectors $V = [V_1 | \dots | V_k]$, where $V_1$ is initialized at random. At iteration $i + 1$, Lanczos performs a matrix-vector product $V_{i+1} \leftarrow G \cdot V_i$, orthogonalizes $V_{i+1}$ against $V_i$ and $V_{i-1}$ and normalizes it. The tridiagonal entries of $T$ are given by coefficients computed during orthogonalization and normalization. See Chapter 1 of Meurant [2006] for a full description of Lanczos algorithm.

**The benefits of Lanczos.**  Lanczos has two features that make it particularly appealing for our uses case. First, Lanczos does not need explicit access to the input matrix $G$, but only access to an implementation of $u \mapsto Gu$. Second, Lanczos uses a small working space: only $3p$ floating point numbers, where the input matrix is $p \times p$. Indeed, we can think of Lanczos as releasing its output in streaming and only storing a tiny state consisting of the last three vectors $V_{i-1}, V_i$ and $V_{i+1}$.

**The curse of numerical instability.**  Unfortunately, the implementation of Lanczos described above is prone to numerical instability, causing $V_1 \dots V_k$ to be far from orthogonal. A careful analysis of

---

[4]The original version statest that the memory footprint is $O(Tp)$, where $T \geq k$ is a parameter of their SVD algorithm.

[5]In this work, we are only concerned with real symmetric matrices.

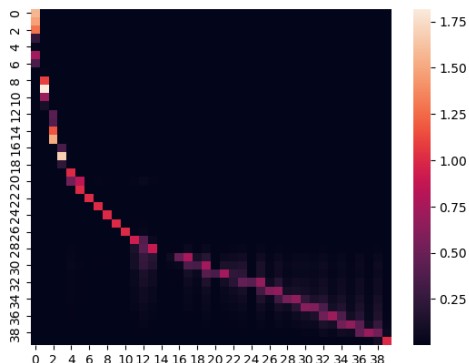

Figure 5: We study the GGN of a LeNet model with $44.000$ parameters trained on MNIST. We run 40 iterations of hi-memory Lanczos and low-memory Lanczos. Let $H = [H_1 | \dots | H_{40}]$, $\Lambda_H$, $L = [L_1 | \dots | L_{40}]$, and $\Lambda_L$ be the eigenvectors and eigenvalues computed by the two algorithms respectively. We sort both sets of eigenvectors in decreasing order of corresponding eigenvalues. In position $(i, j)$ we plot $\langle H_i, L_j \rangle$. It is apparent that multiple eigenvectors $L_j$ correspond to the same eigenvector $H_i$.

the rounding errors causing this pathology was carried out by Paige in his PhD thesis as well as a series of papers Paige [1971, 1976, 1980].

To counteract this, a standard technique is to re-orthogonalize $V_{i+1}$ against all $\{V_j\}_{j \leq i}$, at each iteration. This technique has been employed to compute the low-rank approximation of huge sparse matrices Simon and Zha [2000], as well as in Madras et al. [2019] to compute an approximation to the top-$k$ eigenvectors. Unfortunately, this version of Lanczos loses one the two benefits described above, in that it must store a larger state consisting of the whole matrix $V$. Therefore, we dub this version of thew algorithm hi-memory Lanczos and the cheaper version described above low-memory Lanczos.

Paige Paige [1980] proved that the loss of orthogonality in low-memory Lanczos is strictly linked to the convergence of some of the approximate eigenpairs to the corresponding eigenpairs of $G$. Moreover, the vectors $V_1 \dots V_k$ are not orthogonal because they are tilted towards such eigenpairs. Accordingly, he observed that among the eigenpairs computed by low-memory Lanczos there are multiple eigenpairs approximating the same eigenpair of $G$. This can easily be observed while running both low-memory Lanczos and hi-memory Lanczos and computing the dot products of the their retrieved eigenvectors; see Figure 5.

**Post-hoc orthogonalization.** Based on the observations of Paige Paige [1980], we expect that orthogonalizing the output of low-memory Lanczos post-hoc should yield an orthonormal basis that approximately spans the top-$k$ eigenspace. Since in our method we only need to project vectors onto the top-$k$ eigenspace, this suffices to our purpose.

We confirm this expectation empirically. Indeed, using the same setting of Figure 5 we define $\Pi_{LM}$ as the projection onto the top-10 principal components of $L\Lambda_L$, and define $\Pi_{HM}$ likewise. Then, we measure the operator norm[6] (i.e., the largest singular value) of $\Pi_{LM} - \Pi_{HM}$ and verify that it is fairly low, only $0.03$. Taking principal components was necessary, because there is no clear one-to-one correspondence between $L_i$s and $H_i$s, in that as observed in Figure 5 many $L_i$s can correspond to a single $H_i$. Nonetheless, this proves that low-memory Lanczos is capable of approximating the top eigenspace.

### C.1 Spectral properties of the Hessian /GGN

Spectral property of the GGN and Hessian have been studied both theoretically and empirically. In theory, it was observed that in the limit of infinitely-wide NN, the Hessian is constant through training Jacot et al. [2018]. However, this is no longer true if we consider NN with finite depth Lee et al. [2019]. In Sagun et al. [2017], Papyan [2018], Ghorbani et al. [2019], they show *empirically* that the

---

[6]This can be done efficiently via power method.

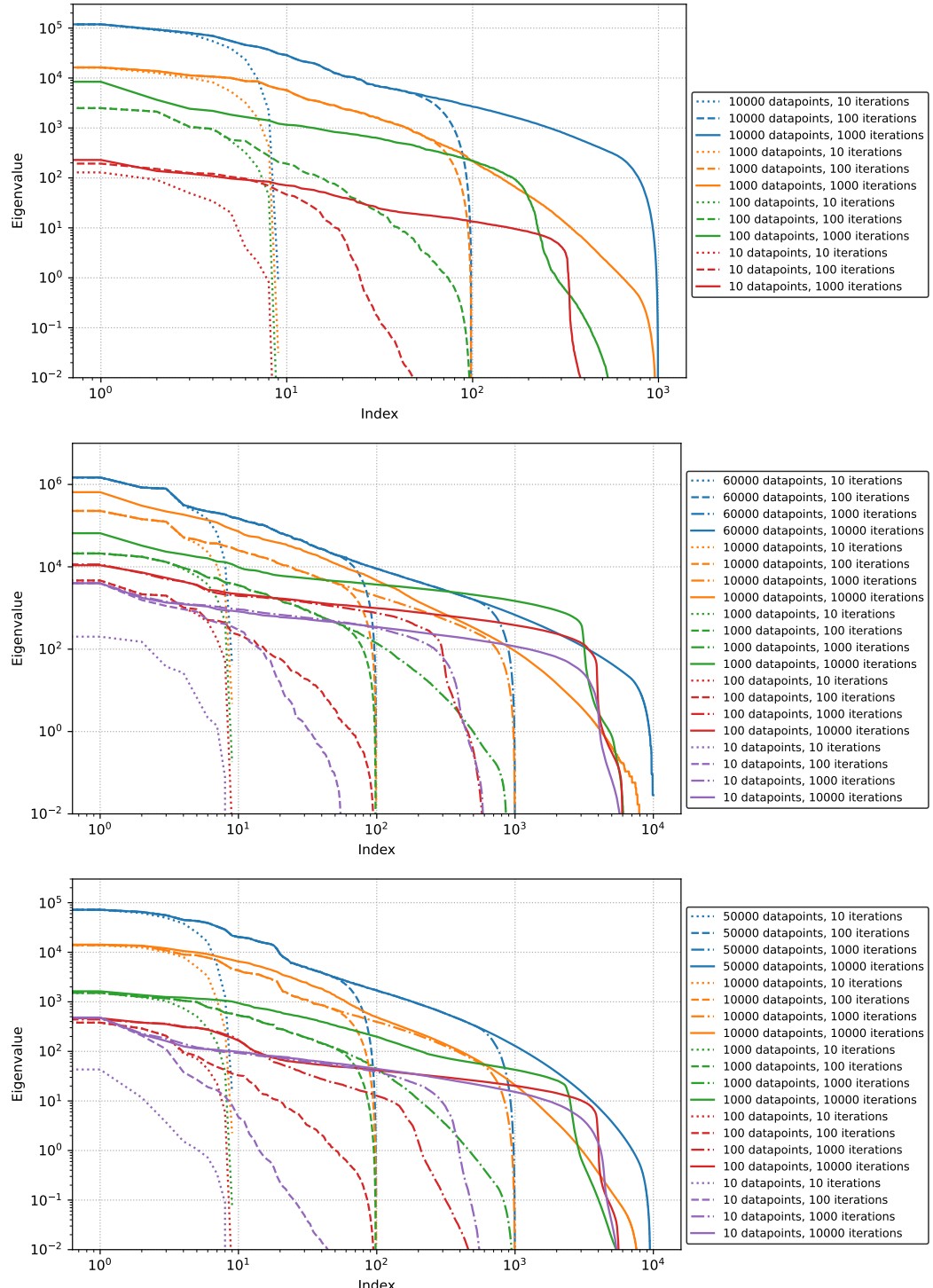

Figure 6: Eigenspectrum obtained from hi-memory Lanczos on: ResNet model ($p = 300$K) trained on CIFAR-10 (top), LeNet model ($p = 40$K) trained on FashionMNIST (middle) and MLP model ($p = 20$K) trained on MNIST (bottom). Standard deviations over 5 Lanczos random seeds.

spectrum of the GGN / Hessian is composed of two components: a bulk of near-zero eigenvalues, and a small fraction of outliers away from the bulk. Moreover, in Ghorbani et al. [2019], they observe a large concentration of the gradients along the eigenspace corresponding to the outlier eigenvalues.

In summary, the spectrum of the GGN / Hessian of a deep NN is expected to decay rapidly, having a few outlier eigenvalues, along which most gradients are oriented. Therefore, we would expect a low-rank approximation of the GGN to capture most relevant information.

We test this phenomenon by using hi-memory Lanczosand performing an ablation over both number of iterations and random initialization of the first vector. Figure 6 shows the results over three different model trained on different datasets. The plots cointains means and std bars over 5 random initializations of the first Lanczos vector. It is interesting to observe how, varying the number of iterations, approximately the first 90% values are the same, while the tail quickly drop down to numerical precision. Based on this observation we choose to only use the top 90% vectors returned by Lanczos.

# D   Experimental setting and extended results

Experiments are reproducible and the code is available at `https://github.com/IlMioFrizzantinoAmabile/uncertainty_quantification`. As explained later in greater details, calling the scripts `train_model.py` and `score_model.py` with the right parameters is enough to reproduce all the numbers we presented.

**Reproducibility.** The script `bash/plot_figure_4.sh` collects the entire pipeline of setup, training, scoring and plotting and finally reproduce Figure 4 and save them in the folder `figures`. The first three plots takes approximately 3 days to complete on a NVIDIA H100 80GB GPU.

The code is implemented in JAX [Bradbury et al., 2018] in order to guarantee max control on the randomness, and, given a trained model and a random seed, the scoring code is fully deterministic (and thus fully reproducible). Nonetheless, the dataloader shuffling used at training time depends on PyTorch (bad) randomness management, and consequently it can be machine-dependent.

## D.1   Training details

All experiments were conducted on models trained on one of the following datasets: MNIST [Lecun et al., 1998], FASHIONMNIST [Xiao et al., 2017], CIFAR-10 [Krizhevsky et al., 2009], CELEBA [Liu et al., 2015] and IMAGENET [Deng et al., 2009]. Specifically we train:

- on MNIST with a MLP consisting of one hidden layer of size 20 with tanh activation functions, we trained with ADAM for 50 epochs with a batch size 128 and a learning rate $10^{-3}$; parameter size is $p = 15910$;

- on FASHIONMNIST with a LeNet consisting of two convolution layers with maxpool followed by 3 fully connected layers with tanh activation functions, we trained with ADAM for 50 epochs with a batch size 128 and a learning rate $10^{-3}$; parameter size is $p = 44426$;

- on CIFAR-10 with a ResNet with $(3, 3, 3)$ blocks of channel sizes $(16, 32, 64)$ with relu activation functions, we trained with SGD for 200 epochs with a batch size 128 and a learning rate 0.1, momentum 0.9 and weight decay $10^{-4}$; parameter size is $p = 272378$;

- on CELEBA with a VisualAttentionNetwork with blocks of depths $(3, 3, 5, 2)$ and embedded dimensions of $(32, 64, 160, 256)$ with relu activation functions, we trained with Adam for 50 epochs with a batch size 128 and a learning rate decreasing from $10^{-3}$ to $10^{-5}$; parameter size is $p = 3858309$.

- on IMAGENET with a SWIN model with an embed dimension of 192, blocks of depths $(2, 2, 18, 2)$ and number of heads $(6, 12, 24, 48)$, we trained with Adam with weight decay for 60 epochs plus a 10 epoch warmup, with a batch size 128 and a learning rate decreasing from $10^{-3}$ to $10^{-5}$; parameter size is $p = 196517106$.

For training and testing the default splits were used. All images were normalized to be in the $[0, 1]$ range. For the CELEBA dataset we scale the loss by the class frequencies in order to avoid imbalances. The cross entropy loss was used as the reconstruction loss in all models but the VisualAttentionNetwork one, on which multiclass binary cross entropy was used.

We train all models with 10 different random seed, (except for ImageNet, that we only trained with one seed) which can be reproduced by running

```bash
bash/setup.sh
source virtualenv/bin/activate
echo "Train all models"
for seed in {1..10}
do
    python train_model.py --dataset MNIST --likelihood classification --model MLP
        --seed $seed --run_name good --default_hyperparams
    python train_model.py --dataset FMNIST --likelihood classification --model
        LeNet --seed $seed --run_name good --default_hyperparams
    python train_model.py --dataset CIFAR-10 --likelihood classification --model
        ResNet --seed $seed --run_name good --default_hyperparams
    python train_model.py --dataset CelebA --likelihood binary_multiclassification
        --model VAN_tiny --seed $seed --run_name good --default_hyperparams
    python train_model.py --dataset ImageNet --likelihood classification --model
        SWIN_large --seed $seed --run_name good --default_hyperparams
done
```

The trained model *test set* accuracies are: $0.952\pm0.001$, $0.886\pm0.003$, $0.911\pm0.003$, $0.889\pm0.002$, $0.672$, respectively for the 5 different settings.

### D.2 Scoring details

The standard deviations of the scores presented are obtained scoring the indepentently trained models. For all methods we use: 10 seeds for MNIST, 10 seeds for FMNIST, 5 seeds for CIFAR-10, 3 seeds for CELEBA and 1 seeds for IMAGENET. The only exception is Deep Ensemble for which we used only 3 independent scores for the budget3 experiment (for a total of 9 models used) and only 1 score for the budget10 experiments (using all the 10 trained model).

For the fixed memory budget experiments we fix the rank $k$ to be equal to the budget (3 or 10) for all baseline, while for our method we were able to have a higher rank thanks to the memory saving induced by sketching, specifically:

- for experiments with MNIST as in-distribution we used a sketch size $s = 1000$ and a rank $k = 45$ for the budget3 and $k = 150$ for the budget10;

- for experiments with FASHIONMNIST as in-distribution we used a sketch size $s = 1000$ and a rank $k = 132$ for the budget3 and $k = 440$ for the budget10;

- for experiments with CIFAR-10 as in-distribution we used a sketch size $s = 10000$ and a rank $k = 81$ for the budget3 and a sketch size $s = 50000$ and a rank $k = 50$ for the budget10;

- for experiments with CELEBA as in-distribution we used a sketch size $s = 10000$ and a rank $k = 100$ both for the budget3 and for the budget10.

- for experiments with IMAGENET as in-distribution we used a sketch size $s = 20000000$ and a rank $k = 30$.

Baseline-specific details:

- SWAG: we set the parameter collecting frequency to be once every epoch; we perform a grid search on momentum and learning rate hyperparameter: momentum in $[0.9, 0.99]$ resulting in the latter being better, and learning rate in $[0.1, 1, 10]$ times the one used in training resulting in 1 being better. Although we spent a decent amount of effort in optimizing these, it is likely that a more fine search may yield slighlty better results;

- SCOD: the truncated randomized SVD has a parameter $T$ and we refer to the paper for full explanation. Importantly to us, the default value is $T = 6k + 4$ and the memory usage at preprocessing is $Tp$, which is greater than $kp$. Nonetheless at query time the memory requirement is $pk$ and we consider only this value in order to present the score in the most fair way possible;

- LOCAL ENSEMBLE: the method presented in the original paper actually makes use of the Hessian, rather than the GGN. And we denoted them as LE-H and LE respectively, which is not fully respectful of the original paper but we think this approach helps clarity. Anyway, we extensively test both variant and the resulting tables show that they perform very similarly;

- DEEP ENSEMBLE: each ensemble consists of either 3 or 10 models, depending on the experiment, each initialized with a different random seed;

- DIAGONAL LAPLACE: has a fixed memory requirement of $1p$.

All the scores relative to the budget3 experiment on CIFAR-10 reported in Table 3 can be reproduced by running the following bash code

```
bash/setup.sh
source ./virtualenv/bin/activate
dataset="CIFAR-10"
ood_datasets="SVHN CIFAR-100 CIFAR-10-C"
model_name="ResNet"
for seed in {1..5}
do
    echo "Sketched Lanczos"
    python score_model.py --ID_dataset $dataset --OOD_datasets $ood_datasets
        --model $model_name --model_seed $seed --run_name good --subsample_trainset
        10000 --lanczos_hm_iter 0 --lanczos_lm_iter 81 --lanczos_seed 1 --sketch
        srft --sketch_size 10000
    echo "Linearized Laplace"
    python score_model.py --ID_dataset $dataset --OOD_datasets $ood_datasets
        --model $model_name --model_seed $seed --run_name good --subsample_trainset
        10000 --lanczos_hm_iter 3 --lanczos_lm_iter 0 --lanczos_seed 1
        --use_eigenvals
    echo "Local Ensembles"
    python score_model.py --ID_dataset $dataset --OOD_datasets $ood_datasets
        --model $model_name --model_seed $seed --run_name good --subsample_trainset
        10000 --lanczos_hm_iter 3 --lanczos_lm_iter 0 --lanczos_seed 1
    echo "Local Ensemble Hessian"
    python score_model.py --ID_dataset $dataset --OOD_datasets $ood_datasets
        --model $model_name --model_seed $seed --run_name good --subsample_trainset
        10000 --lanczos_hm_iter 3 --lanczos_lm_iter 0 --lanczos_seed 1 --use_hessian
    echo "Diag Laplace"
    python score_model.py --ID_dataset $dataset --OOD_datasets $ood_datasets
        --model $model_name --model_seed $seed --run_name good --score diagonal_lla
        --subsample_trainset 10000
    echo "SCOD"
    python score_model.py --ID_dataset $dataset --OOD_datasets $ood_datasets
        --model $model_name --model_seed $seed --run_name good --score scod
        --n_eigenvec_hm 3 --subsample_trainset 10000
    echo "SWAG"
    python score_model.py --ID_dataset $dataset --OOD_datasets $ood_datasets
        --model $model_name --model_seed $seed --run_name good --score swag
        --swag_n_vec 3 --swag_momentum 0.99 --swag_collect_interval 1000
done
for seed in (1,4,7)
    echo "Deep Ensembles"
    python score_model.py --ID_dataset $dataset --OOD_datasets $ood_datasets
        --model $model_name --model_seed $seed --run_name good --score ensemble
        --ensemble_size 3
do
done
```

*Disclaimer*. We did not include the KFAC approximation as a baseline, although its main selling point is the memory efficiency. The reason is that it is a layer-by-layer approximation (and so it neglects correlation between layers) and its implementation is layer-dependent. There exist implementations for both linear layers and convolutions, which makes the method applicable to MLPs and LeNet.

But, to the best of our knowledge, there is no implementation (or even a formal expression) for skip-connections and attention layers, consequently making the method *inapplicable* to ResNet, Visual Transformer, SWIN, or more complex architectures.

*Disclaimer 2*. We did not include the max logit values [Hendrycks and Gimpel, 2016] as a baseline. The reason is that it is only applicable for classification tasks, so for example would not be applicable for CelebA which is a binary multiclassfication task.

**Out-of-Distribution datasets.** In an attempt to evaluate the score performance as fairly as possible, we include a big variety of OoD datasets. For models trained on MNIST we used the rotated versions of MNIST, and similarly for FASHIONMNIST. We also include KMNIST as an MNIST-Out-of-Distribution. For models trained on CIFAR-10 we used CIFAR-100, SVHN and the corrupted versions of CIFAR-10, of the 19 corruption types available we only select and present the 14 types for which at least one method achieve an AUROC$\geq 0.5$. For models trained on CELEBA we used the three subsets of CELEBA corresponding to faces with eyeglasses, mustache or beard. These images were of course excluded from the In-Distribution train and test dataset. Similarly for models trained on IMAGENET we used excluded from the In-Distribution train and test dataset 10 classes (Carbonara, Castle, Flamingo, Lighter, Menu, Odometer, Parachute, Pineapple, Triceratops, Volcano) and use them as OoD datasets. We do not include the results for Carbonara and Menu in thr Table since no method was able to achive an AUROC$\geq 0.5$.

Note that rotations are meaningful only for MNIST and FASHIONMNIST since other datasets will have artificial black padding at the corners which would make the OoD detection much easier.

### D.3 More experiment

Here we present more experimental results. Specifically we perform an ablation on sketch size $s$ in Appendix D.3.1 (on synthetic data) and on preconditioning size in Appendix D.3.2. Then in Appendix D.3.3 we perform again all the experiments in Figure 4, where the budget is fixed to be $3p$, but now with a higher budget of $10p$. We score the 62 pairs ID-OoD presented in the previous section (each pair correspond to a x-position in the plots in Figure 9), with the exception of ImageNet because a single H100 GPU is not enough for it.

### D.3.1 Synthetic data

Here we motivate the claim "the disadvantage of introducing noise through sketching is outweighed by a higher-rank approximation" with a synthetic experiment. For a fixed "parameter size" $p = 10^6$ and a given ground-truth rank $R = 100$ we generate an artificial Fisher matrix $M$. To do so, we sample $R$ uniformly random orthonormal vectors $v_1 \ldots v_R \in \mathbb{R}^p$ and define $M = \sum_i \lambda_i v_i v_i^\top$ for some $\lambda_i > 0$. Doing so allows us to (1) implement $x \mapsto Mx$ without instantiating $M$ and (2) have explicit access to the exact projection vector product so we can measure both the sketch-induded error and the lowrank-approximation-induced error, so we can sum them and observe the trade-off.

For various values of $k$ (on x-axis) and $s$ (on y-axis), we run Skeched Lanczos on $x \mapsto Mx$ for $k$ iteration with a sketch size $s$, and we obtain the sketched low rank approximation $U_S$. To further clarify, on the x-axis we added the column "inf" which refers to the same experiments done without any sketching (thus essentially measuring Lanczos lowrank-approximation-error only) which coincides with the limit of $s \to \infty$. The memory requirement of this "inf" setting is $Pk$, that is the same as the second to last column where $s = P$.

We generate a set of test Jacobians as random unit vectors conditioned on their projection onto $Span(v_1 \ldots v_R)$ having norm $\frac{1}{\sqrt{2}}$. We compute their score both exacltly (as in Equation 9) and sketched (as in Equation 10). In the Figure we show the difference between these two values. As expected, higher rank $k$ leads to lower error, and higher sketch size $s$ leads to lower error. Note that the memory requirement is proportional to the product $ks$, and the figure is in *log-log* scale.

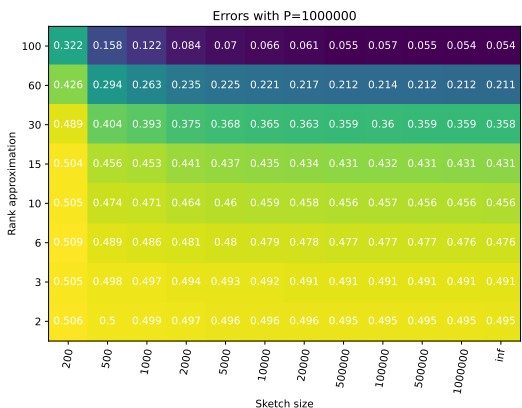

23Figure 7: Ablation on rank $k$ and sketch size $s$.

### D.3.2 Effect of different preconditioning sizes

Figure 8: Preconditioning comparison for LeNet $p = 40$K on FASHIONMNIST vs FASHIONMNIST-rotated-30° (left), and ResNet $p = 300$K on CIFAR-10 vs SVHN (center) or CIFAR-pixelated (right).

Preconditioning can be thought of as a smooth bridge between high-memory Lanczos and SKETCHED LANCZOS. We run the former for a few steps and then continue with the latter on the preconditioned matrix. Consequently, the preconditioning scores "start" from the vanilla Lanczos curve, and improve quicker memory-wise, as shown in Figure 8.

### D.3.3 Budget 10

The results with memory bugdet $10p$ are in Table 4, Table 5 and more extensively in Figure 9.

| | MNIST | | | FASHIONMNIST | |
| | | vs | | | vs |
| | FASHIONMNIST | KMNIST | Rotation (avg) | MNIST | Rotation (avg) |
|---|---|---|---|---|---|
| SLU (us) | $0.28 \pm 0.02$ | $0.46 \pm 0.04$ | $0.61 \pm 0.02$ | $0.95 \pm 0.01$ | $0.75 \pm 0.03$ |
| LLA | $0.27 \pm 0.03$ | $0.34 \pm 0.03$ | $0.52 \pm 0.01$ | $0.88 \pm 0.03$ | $0.72 \pm 0.03$ |
| LLA-D | $\mathbf{0.94 \pm 0.03}$ | $0.98 \pm 0.01$ | $0.73 \pm 0.01$ | $0.68 \pm 0.06$ | $0.61 \pm 0.04$ |
| LE | $0.27 \pm 0.03$ | $0.34 \pm 0.03$ | $0.52 \pm 0.01$ | $0.88 \pm 0.03$ | $0.72 \pm 0.03$ |
| LE-H | $0.27 \pm 0.03$ | $0.34 \pm 0.03$ | $0.52 \pm 0.01$ | $0.88 \pm 0.03$ | $0.72 \pm 0.03$ |
| SCOD | $0.27 \pm 0.02$ | $0.36 \pm 0.03$ | $0.54 \pm 0.01$ | $0.89 \pm 0.02$ | $0.72 \pm 0.03$ |
| SWAG | $0.26 \pm 0.05$ | $0.18 \pm 0.03$ | $0.39 \pm 0.03$ | $0.77 \pm 0.04$ | $0.72 \pm 0.04$ |
| DE | $0.90$ | $\mathbf{1.00}$ | $\mathbf{0.79}$ | $\mathbf{0.99}$ | $\mathbf{0.90}$ |

Table 4: AUROC scores of Sketched Lanczos Uncertainty vs baselines with a memory budget of $10p$.

| | CIFAR-10 | | | CELEBA | |
| | | vs | | | vs |
| | SVHN | CIFAR-100 | Corruption (avg) | FOOD-101 | Hold-out (avg) |
|---|---|---|---|---|---|
| SLU (us) | $\mathbf{0.76 \pm 0.06}$ | $\mathbf{0.54 \pm 0.02}$ | $0.57 \pm 0.04$ | $0.95 \pm 0.003$ | $\mathbf{0.72 \pm 0.02}$ |
| LLA | $0.72 \pm 0.06$ | $0.53 \pm 0.02$ | $0.57 \pm 0.04$ | $0.94 \pm 0.001$ | $0.70 \pm 0.02$ |
| LLA-D | $0.55 \pm 0.10$ | $0.52 \pm 0.02$ | $0.52 \pm 0.04$ | $0.80 \pm 0.02$ | $0.52 \pm 0.03$ |
| LE | $0.72 \pm 0.06$ | $0.53 \pm 0.02$ | $0.57 \pm 0.04$ | $0.94 \pm 0.001$ | $0.70 \pm 0.02$ |
| LE-H | $0.72 \pm 0.07$ | $0.53 \pm 0.02$ | $0.57 \pm 0.04$ | $0.93 \pm 0.002$ | $0.67 \pm 0.02$ |
| SCOD | $0.75 \pm 0.05$ | $\mathbf{0.54 \pm 0.02}$ | $\mathbf{0.58 \pm 0.04}$ | $0.94 \pm 0.002$ | $0.70 \pm 0.02$ |
| SWAG | $0.52 \pm 0.06$ | $0.44 \pm 0.02$ | $0.51 \pm 0.05$ | $0.78 \pm 0.13$ | $0.55 \pm 0.09$ |
| DE | $0.45$ | $0.51$ | $0.50$ | $\mathbf{0.96}$ | $0.68$ |

Table 5: AUROC scores of Sketched Lanczos Uncertainty vs baselines with a memory budget of $10p$.

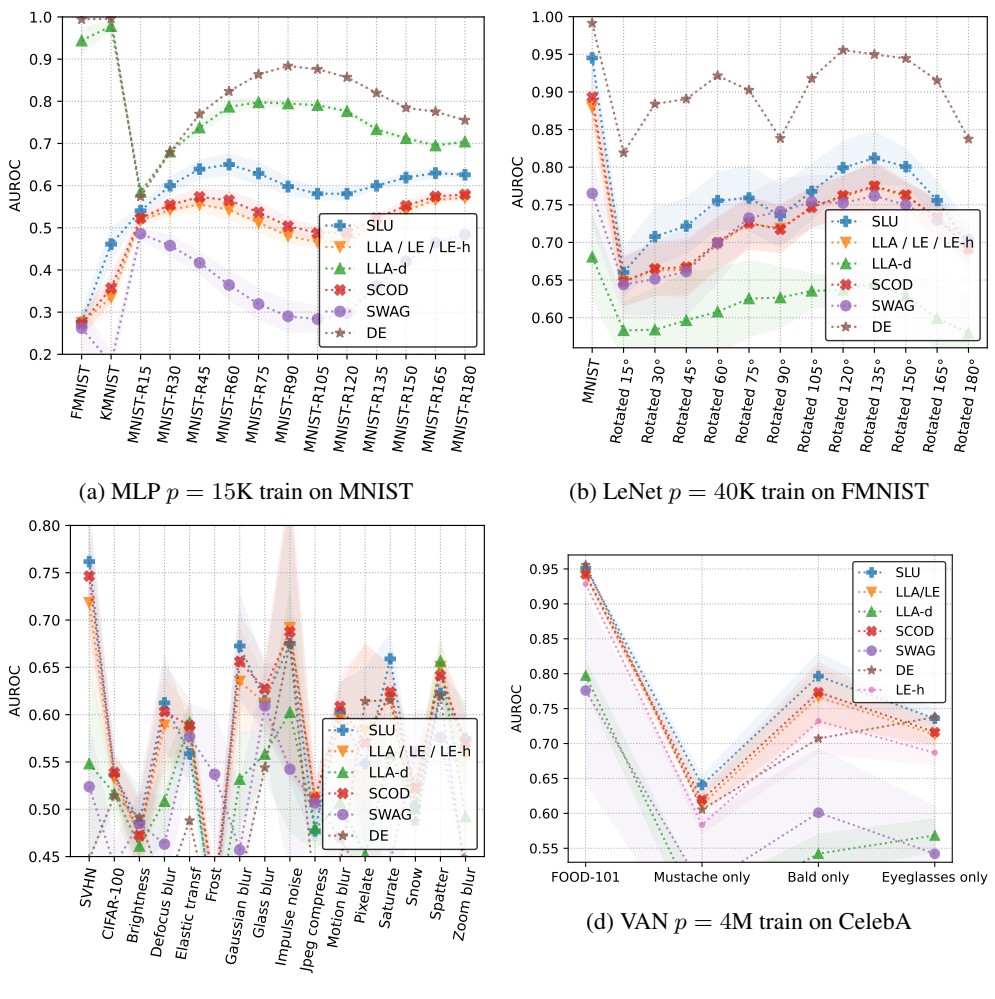

(a) MLP $p = 15K$ train on MNIST

(b) LeNet $p = 40K$ train on FMNIST

(c) ResNet $p = 300K$ train on CIFAR-10

(d) VAN $p = 4M$ train on CelebA

Figure 9: AUROC scores of Sketched Lanczos Uncertainty vs baselines with a memory budget of $10p$.

