# OpenReview forum: "Sketched Lanczos uncertainty score: a low-memory summary of the Fisher information"
_NeurIPS.cc/2024/Conference — NeurIPS 2024 poster_

### Official Review · Reviewer_nMeL · 2024-07-05

**Soundness:** 4
**Presentation:** 3
**Contribution:** 2
**Rating:** 6
**Confidence:** 3

**Summary:**

In this paper, the authors present a memory-efficient way of computing the uncertainty score of a model (the variance of the model's prediction w.r.t. distribution over the model's parameters). Specifically, they combined Lanczos method and sketching to obtain a low-rank approximation of the Generalized Gauss-Newton (GGN) matrix, which is critical in computing the score but has a square size of parameters.

The authors first introduce the Lanczos method since the GGN matrix is more tractable as an operator. However, Lanczos requires re-orthogonalizing the output vectors, leading to extra memory consumption; while the authors' key observation is that the score only depends on the norm of matrix-vector product (Eq.9) which is only slightly influenced by sketching, therefore they can sketch the output on the fly, and conduct orthogonalization and compute the norm in the sketched subspace to save memory. Consequently, the proposed method is called Sketched Lanczos Uncertainty (SLU).

For validation, the authors tested the performance of SLU on out of distribution (OoD) detection tasks with models including MLP, LeNet, and ResNet. Results showed that SLU outperforms existing metrics, achieving a higher AUROC mainly in the low-memory budget scheme (as shown in Fig.3, 4, and Tab.1).

**Strengths:**

Originality: Good. The authors are the first to incorporate sketching with approximate matrix eigendecompositions to reduce the memory footprint.

Quality: Good. The authors provide a comprehensive analysis of their SLU method. For instance, in Fig.2 they illustrated the low-rankness of the GGN matrix and the efficacy of low-rank approximation, also with the core lemma 3.2 they showed the norm can be recovered by the sketched vectors and the sketching matrix.

Clarity: Good. The paper is well-organized for the readers to understand why each feature of SLU is proposed.

Significance: Fair. With SLU the authors compensate the sketching error with a higher-rank approximation and get better OoD detection results than baselines with a low-memory budget. SLU can also be applied to other tasks that require computing the norm of vector products with the U matrix.

**Weaknesses:**

Concerns of this paper are mainly around the experiment part:

1. For the claim that "the disadvantage of introducing noise through sketching is outweighed by a higher-rank approximation" (lines 55~57), the authors should add a simple experiment with synthetic data to verify it.

2. Experiment settings in this paper are too simple; expect to see results with larger-scale models such as Vision Transformers.

**Questions:**

1. Are there other ways of reducing the memory bottleneck in computing uncertainty scores? Since Eq.2 which authors leverage to compute the score is an approximation of the true variance, there may be better alternatives to it. For instance, is there a more computation/memory-efficient form of the covariance matrix $M$ that leads to better scores with less cost?
2. For Tab.1, why do the authors fix the memory budget as $3p$? Looking forward to justifications for this choice.

**Limitations:**

1. The authors only considered the $M_\text{noeig}$ case where singular values are 0/1, need to improve the flexibility of SLU to fit for other $M$s.
2. The proposed method only works well for the low-memory setting. As the number of NNs increases, Sketched Lanczos deteriorates while vanilla hi-memory Lanczos work better. (Fig.3, 4)

---

> ### Author Rebuttal · Authors · 2024-08-07
>
> - "For the claim that "the disadvantage of introducing noise through sketching is outweighed by a higher-rank approximation" (lines 55~57), the authors should add a simple experiment with synthetic data to verify it."
>
> We thank the reviewer for the valuable comment. We run an experiment on synthetic data and the results (Figure 1 in the main rebuttal pdf) confirms our claim. We now proceed to explain the setting.
>
> For a given number of parameters $p$ and a given ground-truth rank $R$ we generate an artificial Fisher matrix $M$. To do so, we sample $R$ uniformly random orthonormal vectors $v_1 \dots v_R\in\mathbb{R}^p$ and define $M = \sum_i \lambda_i v_i v_i^\top$ for some $\lambda_i>0$. Doing so allows us to (1) implement $x \mapsto M x$ without instantiating $M$ and (2) have explicit access to the exact projection vector product so we can measure both the sketch-induded error and the lowrank-approximation-induced error.
>
> For various values of $r$ and $s$, we run Skeched Lanczos on $x \mapsto M x$ for $r$ iteration with a sketch size $s$, and we obtain the sketched low rank approximation $U_S$.
>
> We generate a set of test Jacobians as random unit vectors conditioned on their projection onto $Span(v_1\dots v_R)$ having norm $\frac{1}{\sqrt2}$. We compute their score both exacltly (as in Equation (9)) and sketched (as in Equation (10)). In the Figure we show the difference between these two values.
>
> As expected, higher rank $r$ leads to lower error, and higher sketch size $s$ leads to lower error. Note that the memory requirement is proportional to the product $rs$, and the figure is in log-log scale.
>
>
> To further clarify this results, on the x-axis we added the column "inf" which refers to the same experiments done without any sketching (thus essentially measuring Lanczos lowrank-approximation-error only) which coincides with the limit of $s\rightarrow\infty$. The memory requirement of this "inf" setting is $Pr$, that is the same as the second to last column where $s=P$.
>
> - "Experiment settings in this paper are too simple; expect to see results with larger-scale models such as Vision Transformers."
>
> We have extended the experiments with a 4M parameter Vision Transformer on CelebA dataset. Results are great (much better than the ResNet model included in the submission) and we provide figures and tables in the general rebuttal pdf.
> In the $3p$ budget setting our method clearly outperforms all baselines. In the $10p$ budget Deep Ensemble slightly outperforms in detecting FOOOD101 OoD, but our method still outperform all baselines for the more challenging hold-out-classes OoD datasets.
>
> - "Are there other ways of reducing the memory bottleneck in computing uncertainty scores? Since Eq.2 which authors leverage to compute the score is an approximation of the true variance, there may be better alternatives to it. For instance, is there a more computation/memory-efficient form of the covariance matrix $M$ that leads to better scores with less cost?"
>
> This is a great question and the whole line of research that we reviewed in our related work section aims at answering exactly this question.
>
> There are indeed several works that study memory efficient approximation of the matrix $M$, as referred to in the paper, the most notable are diagonal, block diagonal, and Kronecker factorization. Out of these three, we compare with the first one and outperform it, we didn't include the second one because it requires too much memory and, lastly, we didn't include the third one because it is an architecture-dependent approximation and current implementations are very restricted only to some specific types of network (differently from our model, which is architecture agnostic).
>
> - "For Tab.1, why do the authors fix the memory budget as $3p$? Looking forward to justifications for this choice."
>
> This is an interesting question. Our goal was to show the superiority of SLU in a low-memory regime. Given the number of baselines and ID-OoD dataset pairs, we chose to fix the memory budget in order to display our results more easily.
>
> Then why $3p$ and not $2p$ or $5p$? This was indeed an arbitrary choice. We used both $3p$ and $10p$ (in the Appendix) as we thought this was enough to show the behavior of the method. If the reviewer thinks some different values would be interesting we are happy to run those experiments as well and include them in the paper.
>
> Moreover, we are open to suggestions on experiments without a fixed memory budget and we would be happy to include them.

---

> > ### Comment · Reviewer_nMeL · 2024-08-08
> >
> > Thanks for the great rebuttal. The two supplementary experiments in the attachment addressed my concerns (though the ViT used is still a small model with 4M params), so I increased my rating to 6.

---

> > > ### Author Response · Authors · 2024-08-12
> > >
> > > Regarding scaling to bigger models:
> > > the overhead cost of the sketching operations is neglectable, and we can scale as long as we can perform GGN vector products. This  has essentially twice the cost of a gradient backpropagation, such that the time complexity of computing a rank $k$ approximation is upper bounded by the time complexity of running $2k$ training epochs.
> > >
> > > As a proof of concept we computed the score on CelebA with a **43M** parameters vision transformer in the $3p$ budget setting:
> > > | *score* | FOOD101 | Mustache | Bald  | Eyeglasses |
> > > | ----- | ------- | -------- | ----- | ---------- |
> > > | SLU   | 0.705   | 0.715    | 0.738 | 0.680      |
> > > | LE    | 0.672   | 0.693    | 0.697 | 0.665      |
> > >
> > > We want to emphasise that the model here has not been fine-tune trained and consequently the performance are not ideal. Nonetheless we see the same trend as in the other experiments: sketching improves results without increasing memory use.

---

### Official Review · Reviewer_y4fZ · 2024-07-13

**Soundness:** 3
**Presentation:** 4
**Contribution:** 3
**Rating:** 6
**Confidence:** 4

**Summary:**

This paper provides a new algorithm for approximating the Fisher information matrix. Approximation of the Fisher information matrix is an important task whenever the amount of parameters is very large, such as the case for neural networks. This paper makes use of Lanczos algorithm to find an approximate spectrum of the Fisher information matrix, and particularly targets the low memory setting where such approximations are desirable. In particular, in this paper the authors show that:

-Orthogonalization approximately commutes with sketching, making it feasible to sketch Lanczos vectors on the fly and orthogonalize them afterward.

-Empirically, they demonstrate that in low-memory-budget scenarios, the disadvantage of introducing noise through sketching is outweighed by achieving a higher-rank approximation.

**Strengths:**

This paper is well written and well motivated. Indeed, approximation of the Fisher information is an important task where the parameter count is very high. This paper introduces the regime where their approximation is necessary and provides assistance to the reader for these practical choices throughout the document. The authors detail when Lanczos algorithm is applicable and when it is limited. This naturally motives their algorithm of finding a spanning set of basis vectors for the Kyrlov subspace generated by Lanczos algorithm, and then performing the orthogonalisation. Clearly storing high-dimensional basis vectors is an issue. The authors use embedding techniques to minimise this storage requirement and provide compelling computational examples.

**Weaknesses:**

I do not see significant weaknesses with this paper. I think it would be nice if the choice of embedding procedure was better described in the main text. I believe that this might be a problem dependent choice, and whilst the authors have provided a good default choice, it would be good to know when their choice might give poor results, albeit if this seldom occurs.

**Questions:**

Can the authors provide examples of when the embedding method works/fails. I believe it would be greatly beneficial to the reader if some intuition is shed on when this compression step is a useful computational resource and when it may greatly hinder performance.

**Limitations:**

The authors have not included a limitations section. I think this must be included in a future version and include the dependence of their methods performance on the sketching algorithm used.

---

> ### Author Rebuttal · Authors · 2024-08-07
>
> - "I do not see significant weaknesses with this paper. I think it would be nice if the choice of embedding procedure was better described in the main text. I believe that this might be a problem dependent choice, and whilst the authors have provided a good default choice, it would be good to know when their choice might give poor results, albeit if this seldom occurs."
>
> The embedding is done through multiplication with a sketch matrix. It is true that several choices can be made, an alternative to the \texttt{srft} (used in all the reported experiments) is the \texttt{dense} sketch (Theorem 4, [3]
> ). You can try this alternative with the code provided in the submission by simply changing the command ``--sketch srft" to ``--sketch dense".
>
> The choice of the sketch comes from the theoretical guarantees they offer. Specifically, we require it to be $(1\pm \epsilon)$ oblivious subspace embedding (Definition 2.1) and we require it to be memory efficient. We considered three candidate sketch matrices: SRFT (also named Fast JL transform) [1], Sparse JL transform [2] and Dense JL transform (Theorem 4, [3]). These three sketch matrices give different trade-offs in terms of matrix-vector product evaluation time and memory footprint, which we summarize in the following table.
> Here, $p$ is the original dimension (i.e., the number of parameters), $\varepsilon$ is the approximation parameter for subspace embedding (as in Definition 2.1) and $s$ is the reduced dimension. Let $\omega$ be such that the current best matrix-multiplication algorithm runs in time $n^\omega$.
>
> |           | Time    | Memory |
> | ------- | -------- | ------- |
> | Dense JL | $O(p^\omega)$  | $p^2$   |
> | Sparse JL | $O(p \cdot \varepsilon s)$ | $p \cdot \varepsilon s$    |
> | SRFT | $O(p \log p)$    | $p + s$   |
>
> From the table above, it is clear that SRFT is best if our goal is to minimize memory footprint. At the same time, evaluation time is still quasilinear.
>
> [1] Ailon, Nir, and Bernard Chazelle. "The fast Johnson–Lindenstrauss transform and approximate nearest neighbors." SIAM Journal on computing 39.1 (2009): 302-322.
>
> [2] Kane, Daniel M., and Jelani Nelson. "Sparser johnson-lindenstrauss transforms." Journal of the ACM (JACM) 61.1 (2014): 1-23.
>
> [3] Woodruff, David P. "Sketching as a tool for numerical linear algebra." Foundations and Trends® in Theoretical Computer Science 10.1–2 (2014): 1-157.
>
>
> - "Can the authors provide examples of when the embedding method works/fails. I believe it would be greatly beneficial to the reader if some intuition is shed on when this compression step is a useful computational resource and when it may greatly hinder performance."
>
> The embedding is theoretically guaranteed to work (i.e. to have a small error $\epsilon$ with high probability) as long as the sketch size $s$ is big enough. This means that, as long as this condition is satisfied, there exist no examples where the embedding fails.
>
> To be more explicit, for any choosen sketch size $s$, there exists some $\epsilon$ such that the sketching is guaranteed to not exceed $\epsilon$-error (with high probability). This of course means that for this technique to be of any use, such $\epsilon$ needs to be small compared to the ground truth, and consequently $s$ needs to be ``big enough". If $\epsilon$ is on the same scale of the ground truth, then you essentially measure noise, and the method fails.
>
> For example, in the synthetic data experiment (Figure 1 in the general rebuttal pdf) the ground truth is $~0.7$ (i.e. $1/\sqrt2$). You can see that $s=200$ leads to an error of $0.3$ (so likely practically useless), while $s=5000$ leads to an error of $0.07$ (so likely practically useful).

---

> > ### Comment · Reviewer_y4fZ · 2024-08-09
> >
> > I thank the authors for their response and appreciate their additional comments.
> >
> > I understand that the method is guaranteed to work with $s$ chosen sufficiently large. My question was relating to providing an example where the method may fail, for instance constructing an explicit example in the small $s$ regime. Usually understanding such examples gives insight into points on a methods efficiency. It is unclear to me when I would expect to need to select $s$ to be large, or when I may be okay with a smaller value. My question related to providing some intuition on this point.

---

> ### Author Response · Authors · 2024-08-12
>
> Despite significant investigations, we have been unable to find failure modes beyond those predicted by the theory. Nonetheless, for a **fixed sketch matrix $S$**, one can construct an adversarial example on which the error is huge. One such example is to build a GGN matrix whose eigenvectors are **exactly** orthogonal to the span of the sketch in parameter space (which is an easier task with smaller $s$). This failure, however, depends on the specific $S$, so it will not fail when $S$ is randomized.
>
> Finally it should be mentioned that the sketch size $s$ has to be larger than the rank $k$. If $s$ is smaller then we will attempt to orthogonalize more vectors than the dimensionality of the space they live in, which is guaranteed to fail.

---

### Official Review · Reviewer_Ntg4 · 2024-07-14

**Soundness:** 4
**Presentation:** 3
**Contribution:** 4
**Rating:** 8
**Confidence:** 3

**Summary:**

## Summary

- This paper presents an architecture agnostic technique to compute uncertainty scores for pre-trained neural networks. The space complexity (memory usage) of their proposed technique SLU (Sketched Lanczos Uncertainty) grows logarithmically in the size of model parameters. The experiments reported in the paper use models such as LeNet and ResNet in the range 40K - 300K parameters. This is typically still smaller than pre-trained LLMs which are in millions of parameters or perhaps can be applied to LLMs in LoRA setting.
- More concretely, this paper designs an algorithm to compute the "local ensemble uncertainty estimation" score introduced in Madras et al 2019. https://arxiv.org/pdf/1910.09573
	- This earlier paper quantifies "unreliability" of a prediction on the test-example as "underspecification".
	- They state, "trained model is underspecified at a test input if many different predictions at that input are all equally consistent with the constraints posed by the training data and the learning problem specification (i.e., the model architecture and the loss function)"
	- As a solution they introduce, "local ensembles" , a post-hoc method to measure how much a pre-trained model's prediction is underspecified for a particular test input.
- This paper provides a more memory efficient algorithm to compute this "local ensemble".
- The key observation the authors make in this paper are the following
	- Orthogonalization (used in vanilla Lanczos) approximately commutes with sketching (embedding high dim vectors to lower dim to save on memory). Using this key observation they can change the order of sketching the Lanczos vectors and orthogonalization. Doing orthogonalization post-hoc leads to significant memory-savings.
	- The other results they demonstrate is that in this low-memory budget regime, the disadvantage of introducing noise through sketching is countered by using a higher-rank approximation, leading to better performance.

- Background:
	- **Generalized Gauss Newton Matrix**
		- The local information can be characterized commonly by using the Fisher Information Matrix which is the same as Generalized Gauss Newton Matrix (GGN) Matrix. This GGN matrix is a pxp matrix where p is the number of parameters of the model. Various approximations have been proposed in the literature such as block-diagonal, diagonal etc.
		- In this work, the authors approximate the GGN matrix with a low-rank matrix. The rank-k approximation of the GGN matrix can be computed through two algorithms.
			- the Lanczos algorithm or
			- truncated singular value decomposition algorithm.
		- This is also known as Empirical Fischer matrix (Kutner et al 2019)
	- **Uncertainty Score**
		- The uncertainty at a datapoint x is measured as the variance of the prediction $f(\theta)$ with respect to distribution over parameter $\theta$.
		- For the approaches which only train a single network, various choices of covariance are proposed which are connected to the GGN matrix described above (Kunstner et al 2019)
	- Essentially, the authors in this paper show that low-rank approximation of GGN matrix is a good idea because the eigen values decay exponentially.
	- **Lanczos Algorithm**
		- The lanczos algorithm is an algorithm is an iterative method for tridiagonalizing a symmetric matrix G (pxp).
		- If you stop this algorithm at iteration k, it returns a column orthonormal matrix V (pxk) and a tridiagonal matrix T (kxk) such that $V^T.G.V = T$
		- $V^T.T.V$ approximates the projection of G onto its top k eigenvectors. Projecting G onto its top k eigenspace yields the best rank k approximation of G under any unitarily invariant norm.
		- Once the approximation $V^T.T.V$ of $G$ is available, we can retrieve an approximation of top-k eigenpairs of G by diagonalizing T into $T=W\Lambda W^T$
		- The benefit of Lanczos algorithm is that it only takes 3p space where p is the number of neural network parameters
		- However the downside of Lanczos is that its prone to numerical instability and the $V$ may not be orthogonal. Typically standard implementations re-orthogonalize $V$ after every iteration. This loses the memory benefit of Lanczos.
		- The authors dub this version as hi-memory lanczos.
		- Instead of re-orthogonalizing at every step, an alternative strategy is to store all vectors and orthogonalize at the end. This approach still requires to store all vectors hence is not memory efficient. To make it more memory efficient, the authors propose to use sketching to store these vectors efficiently.
	- **Sketching**
		- Sketching is a technique to store high dimensional vectors to low dimensions so that expected norm of vector dot products is bounded.
		- The authors use Subsampled Random Fourier Transform (SRFT) to obtain the subspace embedding.
- Finally, the authors combine the Lanczos algorithm with Sketching to propose the Sketched Lanczos algorithm.
- The paper also proposes a slight modification of sketched lanczos which first runs hi-memory lanczos for few iterations followed by sketched lanczos iterations. The idea here is that running the high memory lanczos for few iterations results in a better conditioned matrix G'.
- Finally the low rank decomposition of G is used to compute the variance of a prediction, which also gives the uncertainty score of a datapoint.

**Strengths:**

### Strengths

- The paper provides a good overview of various other ways of computing uncertainty and their associated limitations. Background on previous papers on which this work builds upon is also greatly appreciated.
- Experimental results show that sketched lanczos is able to achieve AUROC similar to that of Vanilla lanczos with significantly lesser memory
- The paper is comprehensive, with clear and accessible writing. The experimental results justify the claims made. The literature review provides a solid contextual foundation for the work and the background is thoroughly covered.

**Weaknesses:**

### Weakness

- As I alluded to earlier, the models used in experiments are primarily vision models ranging from 40K params to 300K params, it's not clear whether this approach would work for NLP pre-trained models typically ranging in millions of parameters.
- The datasets used are also primarily small scale vision datasets like CIFAR and MNIST.
- However, I still value in this work and novel insights from authors in optimizing the Lanczos algorithm using Sketching.

**Questions:**

### Questions

- Given that memory consumption of the algorithm is $O(k^2.\epsilon^-2)$. (Line 43) Since this does not depend on the number of parameters. Does this mean this can theoretically be applied to larger models? Also, how does this grow logarithmically in the size of model parameters?
- What is the reason for dip in the AUROC beyond a certain memory cost in Figure 3 and Figure 4?
- What is meant by "number of NNs" ? Do you mean number of neural network parameters?
-

**Limitations:**

The authors have adequately addressed the limitations.

---

> ### Author Rebuttal · Authors · 2024-08-07
>
> - "As I alluded to earlier, the models used in experiments are primarily vision models ranging from 40K params to 300K params, it's not clear whether this approach would work for NLP pre-trained models typically ranging in millions of parameters."
>
> We made additional experiments, specifically, we used a 4M parameter Visual Transformer trained on the CelebA dataset. We also included an extra (easier) OoD dataset: FOOD101. The results are very good and included in the general response pdf. In the $3p$ budget setting our method clearly outperforms all baselines. In the $10p$ budget Deep Ensemble slightly outperforms in detecting FOOOD101 OoD, but our method still outperform all baselines for the more challenging hold-out-classes OoD datasets.
>
> - "As I alluded to
>     The datasets used are also primarily small scale vision datasets like CIFAR and MNIST."
>
> The submission also worked on the CelebA dataset which is not a small-scale vision problem.
> With the updated experiments, we show again favorable performance on CelebA OoD detection with another and more modern architecture (Visual Transformer), in addition to the previously shown favorable performance on ResNet architecture. This experiments suggests that our method works irrespective of the choosen architecture.
>
> - "Given that memory consumption of the algorithm is $O(k^2 \epsilon^{-2})$. (Line 43) Since this does not depend on the number of parameters. Does this mean this can theoretically be applied to larger models? Also, how does this grow logarithmically in the size of model parameters?"
>
>
>  The actual space consumption is $O(k\varepsilon^{-2} \cdot \log p \cdot \log(k / \delta))$ as reported in Lemma 3.1 and 3.2.
> In line 43, we say that the memory consumption of our algorithm scales \emph{essentially} as $O(k^2\varepsilon^{-2})$. In the introduction, we chose to ignore factors $\log p$ and $\log k/\delta$ to make our result more understandable. The word \emph{essentially} suggests that $O(k^2\epsilon^{-2})$ is not exact but conveys the most relevant dependence. Indeed yes, this means this can theoretically be applied to larger models since the $\log p$ term is neglectable even for a billion parameters model.
>
> - "What is the reason for dip in the AUROC beyond a certain memory cost in Figure 3 and Figure 4?"
>
> In a nutshell, a sketch of size $O(k^2 \epsilon^{-2})$ can only handle up to $k$ vectors and guarantee an absolute error $\epsilon$ on our uncertainty score. If more than $k$ vectors are considered, the sketch becomes more noisy, which makes our uncertainty score more noisy and thus performs worse on OoD task.
>
> - "What is meant by "number of NNs" ? Do you mean number of neural network parameters?"
>
> In our figures we measure the memory footprint of our algorithm as the number of trained models (NNs) one could fit in such space, exactly as the size of a Deep Ensemble would. We believe that such a unit of measurement is natural. Indeed, for large models, we expect to dispose of storage comparable with that used for the model itself but not orders of magnitude more than that.

---

### Author Rebuttal · Authors · 2024-08-07

We are grateful for the positive and constructive reviews that we reply to individually below. Some of the individual replies refer to figures and tables that are provided in the PDF attached to this message. These figures provide the requested additional experiments, including demonstrating results on vision transformers and synthetic data experiments.

The additional large-scale experiments are presented in Table 1 and Figure 2. Here we use a $4M$ parameter vision trasformer trained on CelebA, this network is able to train much better compared to the previous ResNet and, consequently, it leads to much better OoD performance. Additionally, given the difficulty of the 3 class-out datasets, we included a new (more out-of-distribution) dataset: FOOD101.

The synthetic data experiment is shown in Figure 1 and it supports the claim ``the disadvantage of introducing noise through sketching [going left on the x-axis] is outweighed by a higher-rank approximation [going up on the y-axis]" that we made in line lines 55-57. Note, for example, the two points $(s,r)=(\textnormal{inf},2)$ and $(s,r)=(20000,100)$ have the same memory requirement. At the same time, they have very different errors: 0.495 and 0.061, respectively.


We hope to have alleviated all concerns and are looking forward to further discussion.

---

### Decision · Program_Chairs · 2024-09-25

**Decision:**

Accept (poster)

**Comment:**

This paper proposes a memory-efficient method for producing uncertainty scores. The main idea is to use a rank-k sketch of the empirical Fisher information matrix, which can be obtained using the Lanczos iteration. The Lanczos iteration is memory-efficient, and the orthogonalization step can be done post-hoc, introducing further memory savings.

The reviews are overall supportive of the paper. Reviewers appreciate the clear motivation, novel method, and the experimental validation of the approach. The concern around the scale of the experiments was largely resolved during the rebuttal phase with additional experiments. Given the positive feedback, I recommend acceptance.